

# Compositional balance should be considered in soil particle-size fractions mapping using hybrid interpolators

Mo Zhang[1,2], Wenjiao Shi[1,3]

[1]Key Laboratory of Land Surface Pattern and Simulation, Institute of Geographic Sciences and Natural Resources Research, Chinese Academy of Sciences, Beijing 100101, China

[2]School of Earth Sciences and Resources, China University of Geosciences, Beijing 100083, China

[3]College of Resources and Environment, University of Chinese Academy of Sciences, Beijing 100049, China

*Correspondence to:* Wenjiao Shi (shiwj@lreis.ac.cn), Institute of Geographic Sciences and Natural Resources Research, Chinese Academy of Sciences. 11A, Datun Road, Chaoyang District, Beijing 100101, China.

**Abstract**. Digital soil mapping of soil particle-size fractions (PSFs) using log-ratio methods has been widely used. As a hybrid interpolator, regression kriging (RK) is an alternative way to improve prediction accuracy. However, there is still a lack of systematic comparison and recommendation when RK was applied for compositional data. Whether performance based on different balances of isometric log-ratio (ILR) transformation is robust. Here, we systematically compared the generalized linear model (GLM), random forest (RF), and their hybrid pattern (RK) using different balances of ILR transformed data of soil PSFs with 29 environmental covariables for prediction of soil PSFs on the upper reaches of the Heihe River Basin. The results showed that RF had better performance with more accurate predictions, but GLM had a more unbiased prediction. For the hybrid interpolators, RK was recommended because it widened data ranges of the prediction results, and modified bias and accuracy for most models, especially for RF. The drawback, however, existed due to the data distributions and model algorithms. Moreover, prediction maps generated from RK demonstrated more details of soil sampling points. Three ILR transformed data based on sequential binary partitions (SBP) made different distributions, and it is not recommended to use the most abundant component of compositions as the first component of permutations. This study can reference spatial simulation of soil PSFs combined with environmental covariables and transformed data at a regional scale.

**Keywords:** soil particle-size fractions; regression kriging; compositional data; isometric log-ratio; generalized linear model; random forest

## 1 Introduction

Recently, spatial interpolation of soil particle-size fractions (PSFs) has become a focus in soil science. More accurate predicted soil PSFs could contribute to a better understanding of hydrological, physical, and environmental processes (Delbari et al., 2011; Ließ et al., 2012; McBratney et al., 2002).

The characteristic of compositional data makes soil PSFs were more impressive than other soil properties. Soil PSFs usually express three components of discrete data – sand, silt, and clay, and carry only relevant percentage information. Soil texture is classified as soil PSFs, which can demonstrate on the ternary diagram. This closure system of the ternary diagram is not



Euclidean space. Instead, it is Aitchison space (so-called the simplex) (Aitchison, 1986). Due to the "spurious correlations"
(Pawlowsky-Glahn, 1984), traditional statistical methods based on the Euclidean geometry may make mistakes when dealing
with soil PSFs data directly (Filzmoser et al., 2009). The constant sum, nonnegative, and unbiased are the key to its spatial
interpolation (Walvoort and de Gruijter, 2001). Data transformation is crucial importance for compositional data to transform
it from the simplex to the real space. Log ratio transformations play a significant role in compositional data analysis, including
additive log-ratio (ALR), centered log-ratio (CLR) (Aitchison, 1986), and isometric log-ratio (ILR) (Egozcue et al., 2003).

7        Currently, though these three log-ratio methods have been widely applied to transform soil PSFs data, different study area

scales and what model use should consider when modeling. For local-scale study areas, geostatistical models combined with
log-ratio transformed data can meet the requirements to map spatial patterns effectively in our previous study (Wang and Shi,
2017). As another perspective, functional compositions combined with the kriging method also applied for soil particle size
curves (PSC) (Menafoglio et al., 2014), which can fully develop the richness of information. It used complete and continuous
information rather than discrete information, and soil PSFs can be extracted from the predicted soil PSCs (Menafoglio et al.,
2016a). Log-ratio transformations can also be combined with functional-compositional data for PSCs' stochastic simulation
(Menafoglio et al., 2016b, Talska et al., 2018). For middle-scale study areas, outliers may lead to the variogram's
overestimation and make prediction errors (Lark, 2000). Therefore, the spatial interpolation should take robust variogram
estimators into account to improve model performance (Lark, 2003). The previous study has already proved that applying
robust variogram estimators in log-ratio co-kriging had significant improvement in mapping performance (Wang and Shi,
2018). For the large-scale study area, geostatistical models limited by soil sampling points and increased spatial variability.
More and more studies have concentrated on mapping soil PSFs using different machine learning models, statistical models
and geostatistical models combined with ancillary data (so-called environmental covariates, EC) on a broad basin scale (Zhang
et al., 2020), national scale (Akpa et al., 2014) and global level (Hengl et al., 2017) using log-ratio transformed data.

22        Among these EC-combined models, linear, machine-learning, geostatistical models, and high accuracy surface modeling

(Yue et al., 2020) have been commonly used in middle-scale or large-scale studies. Linear models such as generalized linear
model (GLM) and multiple linear regression (MLR) have been used in soil PSFs prediction because of flexibility and
interpretability (Lane, 2002; Buchanan et al., 2012). Many of machine-learning models were applied for soil PSFs interpolation
and soil texture classification. For example, tree learners – such as random forest (RF), showed more advantages with abilities
to handle noisy datasets and generated more realistic maps (Zhang et al., 2020). Further, regression kriging (RK) can not only
combine environment covariables by its regression part but also improve model accuracy as a hybrid interpolator for some soil
properties, such as topsoil thickness and pH (Hengl et al., 2004). However, further performance comparison needs to be
conducted in mapping soil compositional data by linear models, machine-learning models, and these models combining with
RK (hybrid patterns).

32        The ILR method performed better in log-ratio methods than ALR and CLR both in theory and in practice (Filzmoser and

Hron, 2009; Wang and Shi, 2018; Zhang et al., 2020). ILR eliminates model collinearity and preserves advantageous properties
such as isometry, scale invariance, and sub-compositional coherence, which is based on orthonormal coordinate systems (so-



called balances) using sequential binary partition (SBP) (Egozcue and Pawlowsky-Glahn, 2005). These choices are not unique.
In other words, multiple sets of ILR transformed data can generate by permutations of components (different SBPs) in
compositional data. The choice of SBPs can be based on prior expert knowledge, using a compositional biplot (Lloyd et al.,
2012) or variograms and cross-variograms (Molayemat et al., 2018). It has been proven in statistical science that different
results were obtained using different choices of SBP balances, and the option of a specific SBP for compositions is crucial for
the intended interpretation of coordinates (Fiserova and Hron, 2011). However, most researchers in soil science ignored this
point. Martins et al. (2016) reported that the clay was taken as the denominator in the ALR method because it was the most
abundant composition components. Few studies have compared the different SBP options from the perspective of accurate
assessment and analyzed whether these differences are due to the general characteristics of specific data sets or log-ratio
transformations.

11       Therefore, based on our previous study, the objectives of this study are to (i) compare the spatial prediction accuracy of soil

PSFs using a generalized linear model (GLM) and random forest (RF) combined with environmental covariables and ILR
transformed data; (ii) determine whether hybrid interpolators (GLMRK and RFRK) can improve the interpolation performance
of GLM and RF; and (iii) explore the distributions of different transformed data and the variation law of precision based on
different choices of SBP balances of ILR.
**2 Methods and materials**
**2.1 Study area**
The study area is the upper reaches of the Heihe River basin (HRB), the Heihe River's birthplace, and the central area of the
runoff generation of the HRB. The elevation is from 1640 m to 5573 m (Fig. 1), and the climate is damp and cold dominated
by the Qilian Mountains. The mean annual rainfall of this study area is 350 mm, and the mean temperature is lower than 4 °C.
Meadow and steppe dominate the vegetation types. Grassland was the primary type of land use, and the main soil classes are
frigid calcic soil in the southwest of this study area. Cold desert soil dominates the southeast, and Castanozems and Sierozems
mainly distribute in the north of the study area.





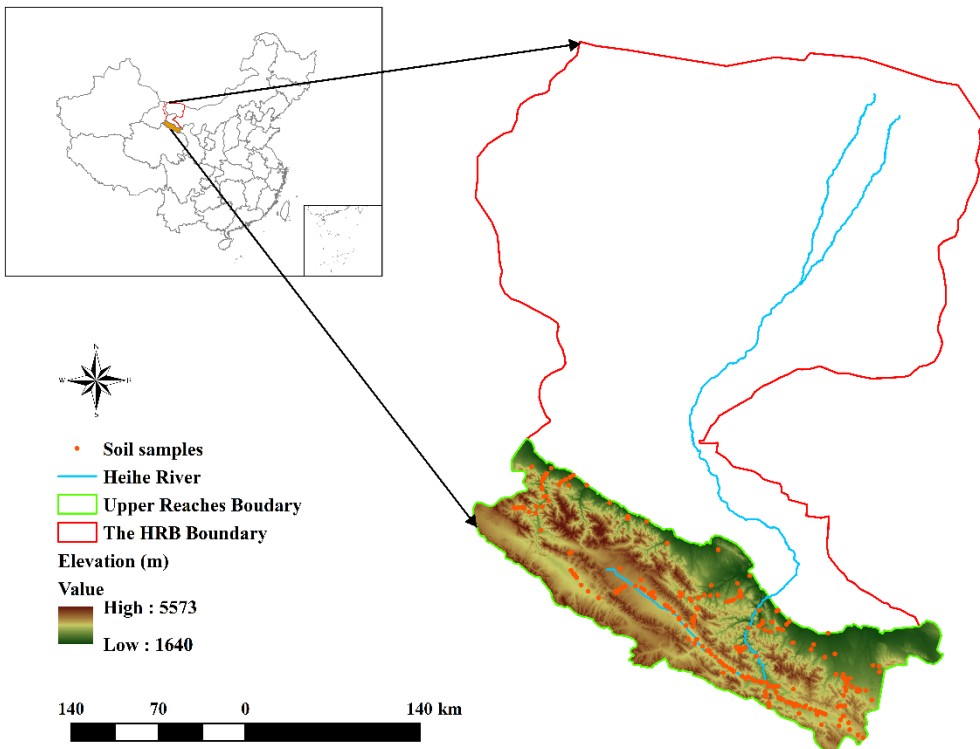

**Figure. 1.** The location, elevation, and soil samples on the upper reaches of the Heihe River Basin.

**2.2 Data collection and analysis**
**2.2.1 Soil PSF data**
A total of 262 soil samples based on a purposive sampling strategy were collected in the upper reaches of the HRB to
characterize the spatial variability of soil PSFs at the regional-scale study area (Fig. 1). The variability of soil formation factors
such as the elevation, soil classes, vegetation classes, and geomorphology classes of the upper reaches of the HRB was
considered in soil samples collection. The average of mixed three topsoil samples (approximately 0 – 20 cm) was obtained to
reduce the noise of soil samples, and the parallel sample was also measured. Subsequently, about 30 g of each soil sample was
air-dried, and the chemical and physical analyses were operated after the fieldwork. Collected soil samples recorded the
information about soil PSFs using Malvern Panalytical Mastersizer 2000 laser with less than 3 % average measurement error.
**2.2.2 The selection of environmental covariables**
There were 29 environmental covariates, including continuous and categorical variables considered in our study (Table 1).
They follow the principles of SCORPAN model (McBratney et al., 2003), which form is defined as $S_a = f(S, C, O, R, P, A, N)$.
$S_a$ are soil attributes (or classes) as a function of soil properties ($S$) or other properties – climatic properties ($C$), organisms
and vegetation ($O$), relief such as topography and landscape attributes ($R$), parent material ($P$), age or time factor ($A$) and



spatial position ($N$). The continuous variables included morphometry and hydrologic characteristics of topographic properties,
climatic and vegetation indices, and soil physical and chemical properties. The categorical variables include geomorphology
types, land use types, and vegetation types, transformed from vector to raster (1000 m). Due to the intricate patterns of
topography in the upper reaches of the HRB, variables of topographic properties dominated the environmental covariates.
SAGA GIS (Conrad et al., 2015) was applied for terrain analysis to derive topographic variables using 30 m DEM (ASTER
GDEM, http://www.gscloud.cn). The collinearity test was used to remove redundant variables, and these topographic
properties were then resampled to 1000 m. More details about environmental covariables can be found in the Data Availability
section.
**Table 1.** Selected environmental covariates in our study.

| Representation | Environment covariables | Abbreviation |
|---|---|---|
| Morphometry characteristics | Analytical Hill Shading | AHS |
| | Aspect | ASPECT |
| | Closed Depressions | CD |
| | Convergence Index | CI |
| | Channel Network Base Level | CNB |
| | Slope Length and Steepness Factor | LSF |
| | Multi-resolution Ridge Top Flatness Index (Gallant and Dowling, 2003) | MRRTF |
| | Multi-resolution Valley Bottom Flatness Index (Gallant and Dowling, 2003) | MRVBF |
| | Mid-slope Position | MSP |
| | Plan Curvature | PLC |
| | Profile Curvature | PRC |
| | Slope Height | SH |
| | Slope Length (D. Moore et al., 1993) | SL |
| | Tangential Curvature (Florinsky, 1998) | TC |
| Hydrologic characteristics | Catchment Area | CA |
| | Surface Area | SA |
| | Stream Power Index | SPI |
| | Topographic Wetness Index (Beven and Kirkby, 1979) | TWI |
| | Vertical Distance to Channel Network | VDCN |
| Climatic and vegetative indices | Average Annual Precipitation | RAIN |
| | Average Annual Temperature | TEM |
| | Normalized Differential Vegetation Index | NDVI |





| Soil physical and chemical properties | Field Water Holding Capacity (Yi et al., 2015; Song et al., 2016; Yang et al., 2016) | FWHC |
|---|---|---|
| | Soil Depth (Yi et al., 2015; Song et al., 2016; Yang et al., 2016) | PDEPTH |
| | Saturated Hydraulic Conductivity (Yi et al., 2015; Song et al., 2016; Yang et al., 2016) | SHC |
| | Soil Organic Carbon | SOC |
| Categorical maps | Geomorphology | GEOT |
| | Land Use | LU |
| | Vegetation Classes | VEGET |

**2.3 Isometric log-ratio transformation and sequential binary partition**
An orthonormal basis of ILR was chosen to project the compositions from $S^D$ (the simplex for the Aitchison geometry) to
$R^{D-1}$ (real space for the Euclidean geometry) isometrically. SBP can be used to explain the choice of a specific orthonormal
basis on $S^D$ with their groups (Egozcue and Pawlowsky-Glahn, 2005). The equation for the choice of construction of
coordinates (so-called balances) between groups of compositions is as follows:
$$z_k = \sqrt{\frac{r_k s_k}{r_k + s_k}} \ln\left(\frac{(x_{i_1} x_{i_2} \dots x_{i_{r_k}})^{1/r_k}}{(x_{j_1} x_{j_2} \dots x_{j_{s_k}})^{1/s_k}}\right), \ k = 1, \dots, D-1, \tag{1}$$
where $z_k$ refers to the balance between two groups, $i_1, i_2, \dots, i_{r_k}$ is the $r_k$ parts of one group, and $j_1, j_2, \dots, j_{r_k}$ is the $s_k$
parts of the other group. Therefore, the balances contain stepwise all the relevant information of compositions in two groups.
It also can be explained in a tabular form – for soil PSFs data (D = 3), all three choices of the balance of SBPs are shown in
Table 2. The first component of ILR contains all the information on soil PSFs, and the main difference of choice of balances
for soil PSFs was the order of three parts, i.e., the first order of soil PSF component was used as numerator of the first ILR
equation. In our study, three balances of SBP — SBP1, SBP2, and SBP3 were transformed from original soil PSF data, and
the orders of soil PSF data were $(sand, silt, clay)$, $(silt, clay, sand)$ and $(clay, sand, silt)$, respectively. The
transformation equation for ILR can be derived from Eq. (1), which was defined as Eq. (2) and Eq. (3). The inverse equations
for ILR were defined as Eq. (4), (5), (6). ILR transformation and its inverse are available in the R package "compositions" (K.
Gerald van den Boogaart and Raimon Tolosana, 2014).
$\mathbf{z} = (z_1, \dots z_{D-1}) = ILR(\mathbf{x})$, and for $i = 1, \dots, D-1$ and component $x_i$, $\tag{2}$
$$z_i = \sqrt{\frac{D-i}{D-i+1}} \ln \frac{x_i}{\sqrt[D-i]{\prod_{j=i+1}^{D} x_j}}. \tag{3}$$
$$Y(x_j) = \sum_{j=1}^{D} \frac{ILR(x_j)}{\sqrt{j \times (j+1)}} - \sqrt{\frac{j-1}{j}} \times ILR(x_j), \tag{4}$$
$ILR(x_0) = ILR(x_D) = 0, \tag{5}$





$$\overline{ILR}(x_j) = \frac{exp(Y(x_j))}{\sum_{j=1}^{D} exp(Y(x_j))}.$$ (6)
**Table 2** All choices of SBPs for soil PSF data (D = 3), the orders of soil PSFs data are $(sand, silt, clay)$, $(silt, clay, sand)$
and $(clay, sand, silt)$ for SBP1, SBP2 and SBP3.

| Groups | Step | Sand | Silt | Clay | r | s | Balance |
|--------|------|------|------|------|---|---|---------|
| SBP1 | 1 | + | - | - | 1 | 2 | Step1: $z_1 = \sqrt{\frac{2}{3}} ln \frac{sand}{\sqrt{silt \times clay}}$ |
| | 2 | 0 | + | - | 1 | 1 | Step2: $z_2 = \sqrt{\frac{1}{2}} ln \frac{silt}{clay}$ |
| SBP2 | 1 | - | + | - | 1 | 2 | Step1: $z_1 = \sqrt{\frac{2}{3}} ln \frac{silt}{\sqrt{clay \times sand}}$ |
| | 2 | - | 0 | + | 1 | 1 | Step2: $z_2 = \sqrt{\frac{1}{2}} ln \frac{clay}{sand}$ |
| SBP3 | 1 | - | - | + | 1 | 2 | Step1: $z_1 = \sqrt{\frac{2}{3}} ln \frac{clay}{\sqrt{sand \times silt}}$ |
| | 2 | + | - | 0 | 1 | 1 | Step2: $z_2 = \sqrt{\frac{1}{2}} ln \frac{sand}{silt}$ |

**2.4 Linear model, machine-learning model, and hybrid patterns**
**2.4.1 Generalized linear model**
The generalized linear model (GLM) is an extended version of the linear model, which contains response variables with non-
normal distributions (Nelder and Wedderburn, 1972). The link function is embedded into the GLM to ensure the classical linear
model assumptions. The scaled dependent variables and the independent variables can be connected using the link function
for the additive combination of model effects. The choice of link function depends on the distribution of response variables
(Venables and Dichmont, 2004). Gaussian distribution with an identity link function was applied in our study, which gives
consequences equivalent to multiple linear regression (Nickel et al., 2014). However, categorical variables can be directly
trained in the GLM without setting dummy variables. The Akaike's information criterion (AIC) was applied to choose the best
predictors and remove model multicollinearity using backward stepwise algorithm.
**2.4.2 Random forest**
Random forest (RF) is a non-parametric technique, which combines the bagging method with a selection of random variables
as an extended version of regression trees (RT) (Breiman, 1996, 2001). It can improve model prediction accuracy by producing
and aggregating multiple tree models. The principle of RF is to merge a group of "weak trees" together to generate a "powerful
forest." The bootstrap sampling method is applied for each tree, and each predictor was selected randomly from all model
predictors. The "out of bag" (OOB) data were applied to produce reliable estimates in an internal validation using a random





subset independent of the training tree data. Three parameters need to be tuned: the number of trees ($ntree$) and minimum size
of terminal nodes ($nodesize$), and the number of variables randomly sampled as predictors for each tree ($mtry$) (Liaw and
Wiener, 2001). The standard value of the parameter for $mtry$ is one-third of the total number of predictors, $ntree$ and
$nodesize$ is 500 and 5, respectively. For regression, the mean square errors (MSEs) of predictions are estimated to train trees.
The variable importance of RF is produced from the OOB data using the "importance" function. The benefits of RFs are that
the ensembles of trees are used without pruning to ensure that the most significant variance can be expressed. Moreover, RF
can reduce model overfitting, and normalization is unnecessary due to the insensitive effects on the value range. The algorithms
of GLM and RF and the parameters adjustment of RF were available in the R package "caret" (Max Kuhn, 2018).
**2.4.3 Regression kriging**
Regression kriging (RK) is a hybrid interpolation technique that combines regression models (e.g., GLM and RF) with ordinary
kriging (OK) of residuals of regression models (Odeh et al., 1995). Mathematically, the RK method corresponds to two
interpolators – the regression part and the kriging part are operated separately (Goovaerts, 1999). A limitation of using only
the regression part is that they are usually only useful within the range of values of the training sets (Hengl et al., 2015). The
RK method's principle is that the regression model explains a deterministic component of spatial variability. The interpolation
of regression residuals generated from OK is used to describe the spatial variability (Bishop and McBratney, 2001; Hengl et
al., 2004). We used residuals to create a variogram (e.g., Gaussian, Spherical, or Exponential) for model based on the MSE
from cross-validation results. Firstly, the regression part in our study (GLM or RF) was used to predict soil PSFs; the residual
from the fitted model was then calculated by subtracting the regression part from the observations. Subsequently, the OK was
applied for the whole study area to interpolate the residuals. Finally, the regression prediction and the predicted residuals at
the same location were summed. The variograms of the RK method were generated automatically by using the
"autofitVariogram" function in the R package "automap" (Hiemstra et al., 2009).
**2.5 Prediction method system and validation**
The method system of spatial interpolation models for soil PSFs in our study was shown in Table 3. We systematically
compared 12 models – four interpolators, including GLM and RF combined with or without RK and three SBPs of ILR
transformation method. For the validation of model performance, the independent data set validation was used to evaluate
models' bias and accuracy. The sub-training sets (70 %) and the sub-testing sets (30 %) were randomly divided from data
independently, and this process was repeated 30 times.
**Table 3.** The method system of spatial interpolation models of soil PSFs.

| Models | GLM | GLMRK | RF | RFRK |
|---|---|---|---|---|
| ILR_SBP1 | GLM_SBP1 | GLMRK_SBP1 | RF_SBP1 | RFRK_SBP1 |
| ILR_SBP2 | GLM_SBP2 | GLMRK_SBP2 | RF_SBP2 | RFRK_SBP2 |




|     | ILR_SBP3 | GLM_SBP3 | GLMRK_SBP3 | RF_SBP3 | RFRK_SBP3 |
|-----|----------|----------|------------|---------|-----------|

2  The mean error (ME), the root mean square error (RMSE), and Aitchison distance (AD) were used to evaluate and compare

3 the prediction performance of models. ME and RMSE measure prediction bias and accuracy, respectively (Odeh et al., 1995).

4 AD is an overall indicator of compositional analysis, which describes the distance between two compositions. Generally, an

5 accurate, unbiased model will have all three symbols close to 0. The equations for ME, RMSE, and AD are defined as:

6 $ME = \frac{1}{n}\sum_{i=1}^{n}(M_i - P_i),$            (7)

7 $RMSE = \sqrt{\frac{1}{n}\sum_{i=1}^{n}(M_i - P_i)^2},$         (8)

8 $AD = \left[\sum_{i=1}^{D}(log\frac{M_i}{G(\boldsymbol{M})} - log\frac{P_i}{G(\boldsymbol{P})})^2\right]^{0.5},$      (9)

9 where $M_i$ and $P_i$ are measured value and predicted value at $i$th position; $n$ refers to the number of soil samples; $D$ is the

10 number of dimensions of compositions; $G(\boldsymbol{M})$ and $G(\boldsymbol{P})$ denotes the geometric mean with the form $G(\mathbf{x}) = (x_1,\ldots,x_D)^{1/D}$

11 of measured and predicted values, respectively.

12 **2.6 Statistical analysis**

13 An interpretation of balances of ILR is based on a decomposition of the covariance structure (Fiserova and Hron, 2011).

14 Therefore, we calculated the variance (VAR), the covariance (COV), and the corresponding correlation coefficient (CC) of

15 ILR transformed data based on different balances of SBP. The VAR, COV, and CC equations are defined as follows, which can

16 derive from Eq (1):

17 $VAR(z) = \frac{1}{r+s}\sum_{p=1}^{r}\sum_{q=1}^{s} var(ln\frac{x_{i_p}}{x_{j_q}}) - \frac{s}{2r(r+s)}\sum_{p=1}^{r}\sum_{q=1}^{r} var(ln\frac{x_{i_p}}{x_{i_q}}) - \frac{r}{2s(r+s)}\sum_{p=1}^{s}\sum_{q=1}^{s} var(ln\frac{x_{j_p}}{x_{j_q}}) -$

18 $\frac{r}{2s(r+s)}\sum_{p=1}^{s}\sum_{q=1}^{s} var(ln\frac{x_{j_p}}{x_{j_q}})$       (10)

19 $COV(z_1, z_2) = \frac{C}{2r_1s_2}\sum_{p=1}^{r_1}\sum_{q=1}^{s_2} var(ln\frac{x_{i_p^1}}{x_{j_q^2}}) + \frac{C}{2r_2s_1}\sum_{p=1}^{r_2}\sum_{q=1}^{s_1} var(ln\frac{x_{i_p^2}}{x_{j_q^1}}) - \frac{C}{2r_1r_2}\sum_{p=1}^{r_1}\sum_{q=1}^{r_2} var(ln\frac{x_{i_p^1}}{x_{i_q^2}}) -$

20 $\frac{C}{2s_1s_2}\sum_{p=1}^{s_1}\sum_{q=1}^{s_2} var(ln\frac{x_{j_p^1}}{x_{j_q^2}}),$       (11)

21 $CC = \frac{COV(z_1, z_2)}{\sqrt{var(z_1)\cdot var(z_2)}}$         (12)

22 For soil PSFs data, Eq. (10), (11) and (12) can be simplified to three dimensions; the relationship between ratios and the

23 dominant roles of ILR transformed data are demonstrated from the covariance structure. All the statistical analyses, such as

24 the descriptive statistics of soil PSFs data, calculation and evaluation of indicators, and the spatial operation of prediction maps,

25 were performed on the R statistical program (R Development Core Team, 2019).



# 3 Results

## 3.1 Exploratory data analysis

### 3.1.1 Descriptive statistics of soil PSFs data

The descriptive statistics of original (raw) and ILR transformed data, silt fraction dominant soil PSFs with more substantial components than those of sand and clay. The distributions of sand and clay fractions were similar (Fig. 2a). ILR transformed data based on three balances of SBP were revealed different distributions (Figs. 2b, 2c, and 2d). For example, two ILR (ILR1, ILR2) components for SBP1 had symmetric distribution around zero value at the $x$-axis (Fig. 2b). In comparison, data generated from SBP2 or SBP3 had to mirror-symmetric deliveries with left-skewed ILR1 of SBP2 and right-skewed ILR2 of SBP3 (Figs. 2c and 2d). The comparison of means and medians demonstrated that back-transformed means of three ILR transformed data were the same, and the mean of sand of ILR was closer to the median compared with soil PSF original data. In contrast, the cases of component silt and clay were the opposite (Fig. 2e).

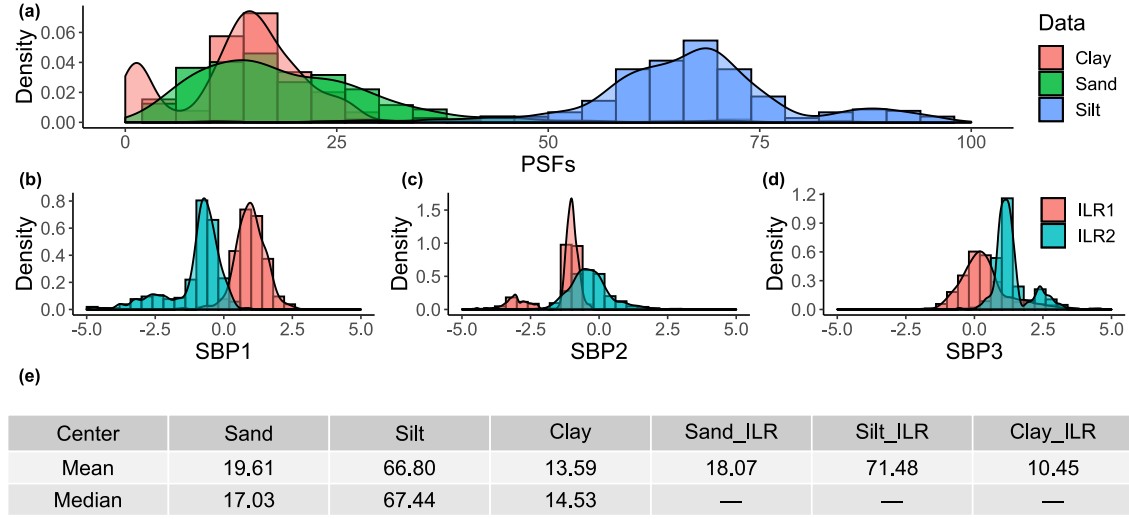

| Center | Sand | Silt | Clay | Sand_ILR | Silt_ILR | Clay_ILR |
|--------|------|------|------|----------|----------|----------|
| Mean | 19.61 | 66.80 | 13.59 | 18.07 | 71.48 | 10.45 |
| Median | 17.03 | 67.44 | 14.53 | — | — | — |

**Figure. 2.** Descriptive statistics of original soil PSF data and ILR transformed data using different balances of SBP. Not that means of Sand_ILR, Silt_ILR, and Clay_ILR from different SBPs of ILR were back-transformed to the real space.

### 3.1.2 Covariance structure of ILR transformed data with different balances

The covariance analysis of the transformed data of soil PSFs data based on different SBPs showed that the variance VarILR_1 of SBP3 was maximum, followed by the values of VarILR_1 of SBP1 and SBP2 (Table 4). The variance of the second component of ILR (VarILR_2) was opposite to the rule of VarILR_1. The covariance (COV) and the corresponding correlation coefficient (CC) followed the same pattern – SBP1 > SBP3 > SBP2. These values revealed the relationship between soil PSFs components or ratios. The first equation of ILR ($z_1$ in Table 2) contained all the information on soil PSFs. The second one ($z_2$ in Table 2) included only two components. Therefore, the information on VarILR_1 was more abundant. Six values of





VarILR_1 and VarILR_2 were not 0 (or not nearly 0), indicating that there was no constant (or almost the constant) in any two
ratios of soil PSF components. The value of COV of SBP3 was close to 0, showing the proportions of *clay/sand* and *clay/silt*
were approximately the same. The same results were generated from the corresponding correlation coefficient (CC).
**Table 4** Covariance analysis of soil PSF data based on different SBPs. VarILR_1 and VarILR_2 denote the variance of the
first and the second component of ILR, respectively. COV refers to the covariance of ILR1 and ILR2. CC is the correlation
coefficient.

| Balances | VarILR_1 | VarILR_2 | COV | CC |
|---|---|---|---|---|
| SBP1 | 0.53 | 0.71 | 0.32 | 0.52 |
| SBP2 | 0.39 | 0.86 | -0.24 | -0.41 |
| SBP3 | 0.94 | 0.30 | -0.09 | -0.16 |

The distribution of soil PSFs sampling data in the ternary diagram (the USDA texture triangle) showed that the main texture
class was silt loam (Fig. 3a). The biplot of soil samples demonstrated that the rays of three components, i.e., sand, silt, and
clay, were reasonably clustered at about 120 ° in three groups (Fig. 3b).

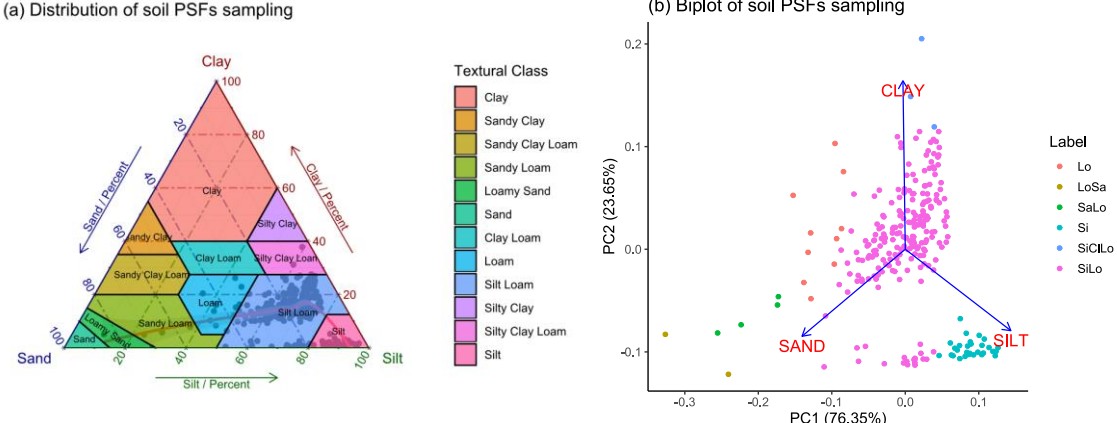

**Figure. 3.** The distribution in the USDA triangle (a) and biplot graph (b) of soil PSFs sampling. The red, smooth curve of these
soil samples in the USDA triangle was fitted by loess function.
**3.2 Accuracy comparison of different models using ILR data**
The first three rows of boxplots (Figs. 4 a, 4b, and 4c) demonstrated the bias of different models according to ME values. The
MEs of sand were closest to 0, followed by MEs of clay and silt. GLM was more unbiased than RF with lower ME values.
After combing with RK, there was an improvement for ME in most GLM and RF models (Figs. 4a, 4b, and 4c). For the
accuracy assessment, RMSEs of silt was higher than the other two components. GLMRK did not perform as well as expect for
RMSEs, which only improved RMSEs of sand component (Fig. 4d). However, RFRK had better performance than GLMRK
and improved most RMSEs of parts compared with RF except for RFRK_SBP1 of sand. Overall indicator of soil PSFs, AD,




showed that RF (or RFRK) performed better than GLM (or GLMRK) in both average RMSE values and uncertainties (Fig.
4g). Moreover, RFRK improved AD values for SBP2 and SBP3 methods. For the uncertainty assessment, RF generated lower
difficulties than GLM, and models combined with RK further reduced the uncertainties for most GLM and RF models. For
three balances of SBP methods, model performances were different. To better evaluate model performance using different SBP
balances, we graded each box from 1 to 3, and the final results were shown in the Supplementary Material table S1.1. It
demonstrated that SBP1 performed best with the lowest ME value among all models. For the accuracy comparison, the pattern
is not apparent, but it can be considered hierarchically. For GLM, SBP1 had better performance than the other two SBPs
methods, which also performed well when RK was added (GLMRK). For RF, SBP1 produced the best result. However, the
introduction of RK made SBP3 performed best among the three methods. Further, the RMSEs generated from RFRK using
SBP3 data had the best accuracy among all models in our study.





**Figure. 4.** Accuracy comparison of GLM, RF, and their RK patterns using different ILR balances data. The mean values of
different model indicators were calculated in their boxes.
**3.3 Spatial prediction maps of soil PSFs generated from different models**
Prediction maps of soil PSFs made from different models were revealed in Figs. 5, 6, and 7. For components of soil PSFs,
three group maps followed a similar rule. GLM and GLMRK showed more extensive ranges of predicted value, and their maps
were more relevant to the real environment. However, RF and RFRK predicted relatively narrow and low values of these
components, revealing smoother than GLMs. Moreover, RK methods demonstrated hot spots and cold spots on prediction
maps compared with only-regression parts. Fig. S2.1 showed more details of soil sampling points.





Figure. 5. Spatial prediction maps of the sand component of the upper reaches of the Heihe River Basin.



**(a) GLM_SBP1**    **(b) GLM_SBP2**    **(c) GLM_SBP3**

**(d) GLMRK_SBP1**   **(e) GLMRK_SBP2**   **(f) GLMRK_SBP3**

**(g) RF_SBP1**    **(h) RF_SBP2**    **(i) RF_SBP3**

**(j) RFRK_SBP1**    **(k) RFRK_SBP2**    **(l) RFRK_SBP3**

Silt (%)

| | | |
|---|---|---|
| 20 - 40 | 50 - 56 | 61 - 65 | 69 - 73 | 77 - 82 |
| 40 - 50 | 56 - 61 | 65 - 69 | 73 - 77 | 82 - 95 |

0 40 80  160  240  320 km

2  **Figure. 6.** Spatial prediction maps of the silt component of the upper reaches of the Heihe River Basin.





**Figure. 7.** Spatial prediction maps of the clay component of the upper reaches of the Heihe River Basin.

**3.4 Spatial distribution of soil texture classes in the USDA triangles**

The predicted soil textures plotted in Fig. 8 in the USDA texture triangles showed that most predicted soil textures fell within the ranges of observed soil textures (Fig. 3a), and silt loam was dominant in the soil texture types for all the cases. GLM produced more discrete distribution than RF, and the RK method expanded the effect of dispersion. For the trends of predicted samples, silt components predicted from all models were over-estimated. The pattern fitting curves indicated that the prediction





results were closer to the values in the bottom right of the USDA soil texture trianglethan the soil PSFs observations. Curves
of GLMRK and RFRK were longer than GLM and RF, showing more extensive ranges of value in ternary. Compared with
GLMRK, RFRK produced more upward extension (Fig. 8j, k, l). It was clear that the clay fraction was over-estimated, and the
sand fraction was under-estimated.







**Figure. 8.** Predicted 262 soil samples based on leave-one-out method in USDA texture triangles using (a) GLM_SBP1, (b) GLM_SBP2, (c) GLM_SBP3, (d) GLMRK_SBP1, (e) GLMRK_SBP2, (f) GLMRK_SBP3, (g) RF_SBP1, (h) RF_SBP2, (i) RF_SBP3, (j) RFRK_SBP1, (k) RFRK_SBP2, (l) RFRK_SBP3. Red fitting lines in triangles showed the trends.



## 4 Discussion

### 4.1 Comparison of GLM, RF and their hybrid interpolators using ILR data

For the assessment of independent validation, RF revealed more accurate but more bias than GLM. RK improved the performance of the bias for most models and the accuracy of RF. Odeh et al. (1995) have indicated that RK was superior to the linear models like MLR in sand prediction. Scarpone et al. (2016) reported that RFRK outperformed RF when dealing with soil thickness prediction as a hybrid interpolator. We proved that RK was also available for compositional data to improve performance using ILR transformation in RF. In summary, GLM and RF had their advantages and disadvantages when considering the trade-off between bias and accuracy. The difficulty of GLM is back-transformation; it needs to present results on the original untransformed scale after analyzing on a transformed level, which may produce the unfortunate result between them (Lane, 2002). In our study, we compared the means of ILR transformed data and the original data. We proved the feasibility of the ILR transformation method, especially for meeting the requirements of compositional data. Still, the accuracy of GLM needs to be improved; this may be because the transformed data did not follow a normal distribution. In addition, although RF had an advantage on prediction accuracy, the limited interpretability of the consequences – a "black box" effect – made it challenging to modify prediction bias because each tree from the model cannot be examined individually (Grimm et al., 2008). ILR transformation before modeling increased the difficulty of interpretation for the predicted values on ILR-scale and the residuals. Moreover, the back-transformation of the optimal estimate of log-ratio variables does not generate the optimal estimation of compositions (Lark and Bishop, 2007), which also should be considered.

### 4.2 Comparison of three balances of ILR transformation method

The comparison of three balances of SBP showed that most indicators of ME and RMSE using SBP1 of ILR transformed data performed better, which may be interpreted as the distributions of the ILR1 and ILR2 of SBP1 were more symmetric (Fig. 2b). In contrast, the performance of SBP2 was worse than the other two because the ILR_1 component, including all the information of soil PSFs, was left-skewed (Fig. 2c). This result was apparent, especially for GLM and GLMRK, because the normal distribution of data is needed in the linear model (Lane, 2002).

The interpretation of the negligible difference among three balances of SBP was the biplot of soil PSFs sampling data (Fig. 3b), which revealed a triangular shape. In other words, three soil PSFs had a mixed pattern, and each component was dominated by the components in one cluster (Tolosana-Delgado et al., 2005). Although the silt component dominated the soil PSFs with the highest content (Fig. 2a), sand and clay played essential roles in compositional data. Therefore, taking the most abundant composition components as the denominator (Martins et al., 2016) or the first component of permutations was not convincing evidence. In contrast, using the most abundant component of compositions as the primary component of alterations, i.e., SBP2, demonstrated relatively poor performance among three SBPs data. Thus, we recommended using other parts that were not the most abundant as the first component of permutations when the biplot diagram was uniform distribution with 120 ° (Fig. 3b). Furthermore, the choice of balance is the key to improving model accuracy, such as the RFRK-SBP3 model (Fig. 4). We also



fitted the biplots using a random sampling test (70 % soil sampling data randomly), and the distribution of these graphs (angle)
were almost the same (Fig. S3.1). Multiple data sets should be considered in further researches to verify if it was a general
feature of soil PSFs samples or produced from our data set.

4         Also, the weighting problem was not considered in this study, because the ILR method can be qualified as an unweighted

log-ratio transformation, giving all parts the same weight for both the definition of the total variance and the reduction of
dimension. It may enlarge the ratios generated from the rare parts and dominate the analysis (Greenacre and Lewi, 2009). The
pairwise log-ratio can set weights by their proportions when there is no additional knowledge about the component
measurement errors (Greenacre, 2019). Nevertheless, all three parts of soil PSF data dominate the biplot diagram without the
rare element and redundancy; thus, there are no shortcomings mentioned above. The accuracy assessments using pairwise log-
ratio transformation need more researches in the future.
**5 Conclusions**
We evaluated and compared the performance of GLM, RF, and their hybrid pattern (i.e., GLMRK and RFRK) using different
ILR balances transformed data. The bias of GLM was lower than those of RF; however, the accuracy of GLM was relatively
lower. GLMs produced more discrete distributions and broader ranges of prediction value distributions in the USDA soil
texture triangles. In other words, GLM and RF generated different data sets – unbiased and inaccurate predictions for GLM
and biased and more accurate predictions for RF.

17        The hybrid patterns of GLM and RF, i.e., RK, were recommended, which produced relative higher prediction accuracy and

environmental correlation, showing more details about soil sampling points (hot spots and cold spots) than the regression part.
However, the non-normal distribution of ILR transformed data, and the "black box" effect of the RF algorithm were drawbacks
of GLMRK and RFRK.

21        Concerning different balances of SBP, three SBP-based data generated different distributions. Their prediction results had a

slight difference. The pattern was not obvious, which was because the angle of the biplot diagram – three rays of soil PSFs
components clustered into three modes, and each part dominated in its cluster. Using the most abundant component of
compositions as the first component of permutations was not the right choice because of the worst performance of SBP2. On
the contrary, we recommended using other parts that were not the most abundant as the first component of permutations when
the biplot diagram was uniform distribution with 120 °, like our study. For a general feature of soil PSFs compositional data,
multiple soil PSFs data sets should be considered and compared in the future. This study reference spatial simulation of soil
PSFs combined with environmental covariables at a regional scale and how to choose the ILR balances.

***Data Availability.*** We did not use any new data and the data we used come from previously published sources. Soil particle-
size fractions data is available through our previous studies (Wang and Shi, 2017, 2018). Moreover, it also can be visited on
this website: http://data.tpdc.ac.cn/zh-hans/data/7f91d36d-8bbd-40d5-8eaf-7c035e742f40/ (Digital soil mapping dataset of
soil texture (soil particle-size fractions) in the upstream of the Heihe river basin (2012-2016); last access: 4 July 2020). The



meteorological data can be accessed through http://data.cma.cn/ (last access: 4 July 2020). Environmental covariates data of
soil physical and chemical properties and categorical maps can be obtained through http://data.tpdc.ac.cn/zh-hans/ (last access:
4 July 2020), including saturated water content, field water holding capacity, wilt water content, saturated hydraulic
conductivity data (http://data.tpdc.ac.cn/zh-hans/data/e977f5e8-972b-42a5-bffe-cd0195f3b42b/, Digital soil mapping dataset
of hydrological parameters in the Heihe River Basin (2012); last access: 4 July 2020), and soil thickness data
(http://data.tpdc.ac.cn/zh-hans/data/fc84083e-8c66-4a42-b729-4f19334d0d67/, Digital soil mapping dataset of soil depth in
the Heihe River Basin (2012-2014); last access: 4 July 2020). DEM data set is provided by the Geospatial Data Cloud site,
Computer Network Information Center, Chinese Academy of Sciences. (http://www.gscloud.cn, last access: 4 July 2020).
***Author contribution.*** Wenjiao Shi contributed to soil data sampling, oversaw the design of the entire project. Mo Zhang
performed the model analysis and wrote the manuscript. Both authors contributed to writing this paper and interpreting data.
***Competing interests.*** The authors declare that they have no conflict of interest.
***Acknowledgment.*** Our team expresses gratitude to the following institutions, Key Laboratory of Land Surface Pattern and
Simulation, Institute of Geographic Sciences and Natural Resources Research, Chinese Academy of Sciences; School of Earth
Sciences and Resources, China University of Geosciences; College of Resources and Environment, University of Chinese
Academy of Sciences. This study was supported by the National Key Research and Development Program of China (No.
2017YFA0604703), the National Natural Science Foundation of China (Grant No. 41771111 and 41771364), Fund for
Excellent Young Talents in Institute of Geographic Sciences and Natural Resources Research, Chinese Academy of Sciences
(2016RC201), and the Youth Innovation Promotion Association, CAS (No. 2018071).

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
