# Peer review of "Compositional balance should be considered in soil particle-size fractions mapping using hybrid interpolators"

_Hydrology and Earth System Sciences, 2020_

## Referee Comment (RC1) · Anonymous Referee #1 · 30 Sep 2020

This paper presents an investigation on the use of compositional balances for the analysis and spatial mapping of particle-size fractions. In particular, it aims to compare the use of ILR balances for the purpose of performing: (1) linear regression (named GLM, although it is just LM in the study), (2) regression kriging, (3) random forest prediction.

I believe that the general topic of the investigation could be interesting for the applied field of study. However, in my view, the study has severe limitations, and should not be considered further for publication in this journal. The main weak points of the study are summarized in the following points.

1. Although only partially explained, the methods being compared appears to be all

applied separately to the three univariate ILR coordinates. This is not correct, as the ILR coordinates are typically correlated, so the analysis should also consider the cross-correlations among coordinates. All the presented results are thus suboptimal, and does not provide any effective guidance for other studies in the field. This also leads inconsistencies in the results (see my following point 2.).

2. The results of linear models and kriging, if correctly applied, should not depend on the ILR basis being chosen. This has been proved by previous studies on the use of linear models and kriging for compositional data (see, e.g., Pawlowsky-Glahn et al, (2015)). The authors cited in the text (Fiserova & Hron, 2011) indeed suggest to choose the ILR basis driven by interpretation purposes (which may be eased by a particular basis), but they do not refer to the influence of the basis on the results themselves, as these are independent on the basis being chosen if the method can be restated in term of a projection in the simplex (as LM and RK). As such, studying the effect of the choice of the ILR basis in these cases does not provide any meaningful information, beside the evidence that the methods discussed in the manuscript were not applied correctly (see point 1.).

3. Studying the bias of linear models and regression kriging is not meaningful, because both are unbiased methods. If bias is found, it derives from an incorrect definition of the notion of bias, which should be considered in the geometry of the simplex. More-over, all the statistics and summaries should be considered in a multivariate setting, and the consideration of univariate components of psfs should be completely avoided (particularly if the aim is to approach them in a compositional setting). Overall, the paper does not discuss clearly the background on compositional data analysis, and the comments to analyses and results are often formally inappropriate, showing inconsistencies and general confusion on the concepts related with the theory of compositional data analysis.

4. The discussion is very confused, and the overall message strongly hindered by incorrect English wording.

References: • Fišerová E, Hron K (2011) On the interpretation of orthonormal coordinates for compositional data. Mathematical Geosciences 43(4):455–468 • Pawlowsky-Glahn V, Egozcue JJ, Tolosana-Delgado R (2015) Modeling and analysis of compositional data. John Wiley & Sons

———————————————————

---

## Referee Comment (RC2) · Anonymous Referee #2 · 20 Oct 2020

The presented manuscript proposes a comparison, in the context of the use of compositional data analysis (CoDA) techniques to perform digital soil mapping of particle-sized fractions, of different ILR transformation choices and different prediction algorithms.

The authors, after having provided a brief analysis of the current literature on the use of compositional data in geosciences, they perform three different ILR transformations of the data, and then proceed to assess sat comparing the prediction accuracy of several statistical learning methods, namely linear regression (glm with gaussian errors and identity link is classical least squares, gaussian regression), universal kriging and

random forests, via the use of a real world dataset.

They then conclude by assessing what is the best algorithm in terms of prediciton by inspecting several performance metrics.

While I do think that the general topic of investigation is of quite interest for an audience of geosciences practitioners, and so it is coeherent with the aims of this Journal, I am quite concerned by the execution of the paper, and I think some very serious points need to be tackled before this paper is able to be considered suitable for publication:

1. The wording is very obscure at times, hindering the very comprehension of the matters at hand 2. Judging by how the performance metrics are chosen, the prediciton problems solve by the authors are all scalar ones, and so the methods seem to having been applied separately to the different components. This is wrong, as it is fundamental in a compositional setting to inspect the cross-correlations between variables (and thus use multivariate prediction methods) 3. Given that linear methods (such as linear regression and regression kriging) are invariant to the choice of ILR basis, I am baffled by seeing results for this methods that are different across different ILR transformation. 4. The estimation of a bias metric via the use of RMSE on unbiased estimators (such as LM and RK) is simply incorrect.

---

## Author Comment (AC1) · 24 Nov 2020

Dear Referee, on behalf of my co-authors, we thank you very much for giving us an opportunity to revise our manuscript. Our manuscript has been reviewed. We appreciate the editor and reviewers very much for their positive and constructive comments and suggestions on our manuscript entitled "Compositional balance should be considered in soil particle-size fractions mapping using hybrid interpolators" (ID: hess-2020-384). We have studied the referee's comments carefully and we have tried our best to revise our manuscript according to the comments and suggestions. Attached please find the revised version and the response to the referee, which we

would like to resubmit for your kind consideration. We are looking forward to your response soon. Thank you and best regards.

Please also note the supplement to this comment:
https://hess.copernicus.org/preprints/hess-2020-384/hess-2020-384-AC1-supplement.pdf

────────────────────────

**Supplement:**

**Response**

    Comments to the Author:

This paper presents an investigation on the use of compositional balances for the analysis and spatial mapping of particle-
size fractions. In particular, it aims to compare the use of ILR balances for the purpose of performing: (1) linear regression
(named GLM, although it is just LM in the study), (2) regression kriging, (3) random forest prediction. I believe that the general
topic of the investigation could be interesting for the applied field of study. However, in my view, the study has severe
limitations, and should not be considered further for publication in this journal. The main weak points of the study are
summarized in the following points.

**Comment 1.** Although only partially explained, the methods being compared appears to be all applied separately to the three
univariate ILR coordinates. This is not correct, as the ILR coordinates are typically correlated, so the analysis should also
consider the cross correlations among coordinates. All the presented results are thus suboptimal, and does not provide any
effective guidance for other studies in the field. This also leads inconsistencies in the results (see my following point 2.).

**Response:** Although our models were predicted separately for each ILR component (ILR1 and ILR2) and seem to be
suboptimal, we think that correlations among the components (i.e., sand, silt, and clay) can be revealed using an ILR
transformation. Therefore, the models considered the joint fractions by transforming the original soil PSF data from simplex
(three components) to the real space (two ILR components). Moreover, the reason why we predicted each ILR component
separately is because that was a more suitable approach for the spatial prediction models currently used (such as the GLM and
RF). In general, in the formula for a single prediction model (GLM and RF), only one column of observations (ILR1 or ILR2)
is included, generating one column of predictions. Therefore, these models cannot consider multiple variables (observations,
ILR1 and ILR2) together in one formula. Some previous studies (Akpa et al., 2014; Buchanan et al., 2012; Huang et al., 2014;
Nagra et al., 2017) have used similar methods in combination a with log-ratio transformation to make predictions of soil PSF
in other study areas, and we think our results can therefore provide guidance for other studies. For the multivariable methods,
we have used compositional kriging for the spatial prediction of soil PSFs in our previous studies (Wang and Shi, 2017, 2018);
however, this approach cannot be combined with environmental covariables to achieve one of the objectives of this work, i.e.,
using hybrid interpolation. For the other models, a multivariate RF may be an alternative method for considering multivariate
settings in future research. We have improved this part of the paper in the revised version (Discussion **4.3 Limitations**)

**P21L549: 4.3 Limitations.**

*"In this work, we used ILR transformation to demonstrate the correlation of soil PSF data, and different balances were also*
*compared. However, these models were predicted separately for each ILR component (ILR1 and ILR2), which were suboptimal*
*because they cannot further consider the cross correlations among ILR coordinates. In our pervious study, we have used*
*compositional kriging (CK) for the spatial prediction of soil PSFs (Wang and Shi, 2017), and the cross correlations of ILRs*
*can be taken into account using CK. Although it is optimal, it cannot consider different balances of ILR, nor can it be combined*

*with the hybrid interpolator (e.g., RK). Moreover, predicting each ILR component separately was a more suitable approach*

*for the spatial prediction models currently used (such as the GLM and RF). Therefore, more alternative spatial prediction*

*models combined with interpretation of ILR balances for compositional data should be considered in the future. For example,*

*CK and high accuracy surface modelling (HASM; Yue et al., 2016) can be applied for small scale study areas. For large scale*

*study areas, multivariate RF (Segal and Xiao, 2011) can be combined with a log-ratio transformation and hybrid interpolation,*

*enabling the cross correlations among ILR coordinates to be better interpreted."*

**Refrence**

Akpa, S. I. C., Odeh, I. O. A., Bishop, T. F. A., and Hartemink, A. E.: Digital Mapping of Soil Particle-Size Fractions for

Nigeria, Soil Sci. Soc. Am. J., 78, 1953-1966, 10.2136/sssaj2014.05.0202, 2014.

Buchanan, S., Triantafilis, J., Odeh, I. O. A., and Subansinghe, R.: Digital soil mapping of compositional particle-size fractions using proximal and remotely sensed ancillary data, Geophysics, 77, WB201-WB211, 10.1190/geo2012-0053.1, 2012.

Huang, J., Subasinghe, R., and Triantafilis, J.: Mapping Particle-Size Fractions as a Composition Using Additive Log-Ratio

Transformation and Ancillary Data, Soil Sci. Soc. Am. J., 78, 1967-1976, 10.2136/sssaj2014.05.0215, 2014.

Nagra, G., Burkett, D., Huang, J., Ward, C., and Triantafilis, J.: Field level digital mapping of soil mineralogy using proximal and remote-sensed data, Soil Use Manage., 33, 425-436, 10.1111/sum.12353, 2017.

Segal, M. and Xiao, Y. Y.: Multivariate random forests, Wiley Interdisciplinary Reviews-Data Mining and Knowledge

Discovery, 1, 80–87, https://doi.org/10.1002/widm.12, 2011.

Yue, T., Liu, Y., Zhao, M., Du, Z., and Zhao, N.: A fundamental theorem of Earth's surface modelling, Environ. Earth Sci.,

75, 751, https://doi.org/10.1007/s12665-016-5310-5, 2016.

Wang, Z., and Shi, W. J.: Mapping soil particle-size fractions: A comparison of compositional kriging and log-ratio kriging,

J. Hydrol., 546, 526-541, 10.1016/j.jhydrol.2017.01.029, 2017.

Wang, Z. and Shi, W.: Robust variogram estimation combined with isometric log-ratio transformation for improved accuracy of soil particle-size fraction mapping, Geoderma, 324, 56–66, https://doi.org/10.1016/j.geoderma.2018.03.007, 2018.

**Comment 2.** The results of linear models and kriging, if correctly applied, should not depend on the ILR basis being chosen.

This has been proved by previous studies on the use of linear models and kriging for compositional data (see, e.g., Pawlowsky-

Glahn et al, (2015)). The authors cited in the text (Fiserova & Hron, 2011) indeed suggest to choose the ILR basis driven by interpretation purposes (which may be eased by a particular basis), but they do not refer to the influence of the basis on the results themselves, as these are independent on the basis being chosen if the method can be restated in term of a projection in the simplex (as LM and RK). As such, studying the effect of the choice of the ILR basis in these cases does not provide any meaningful information, beside the evidence that the methods discussed in the manuscript were not applied correctly (see point

1.).

**Response:**

For the same soil sampling point within the soil PSF raw data, different ILR balances produced different ILR values (ILR1
and ILR2). There is no doubt that they can be back-transformed with the same values of soil PSFs (sand, silt, and clay) even
though the balances were different (Fig. 1). For the soil PSF interpolation, the raw data first transformed the ILR mode (two
components of ILR1 and ILR2), then interpolated and finally back-transformed to the raw data form (three components of
sand, silt, and clay). Using three SBPs resulted in different input values for the interpolation, and also produced different results.
Therefore, for soil PSFs the ILR balance should be selected carefully.

[Figure]

**Fig. 1.** Transformation and inverse transformation of ILR methods based on different SBPs.
Different GLM and GLMRK models based on three ILR balances generated different results in our study, but this is not
indicating that choosing the ILR basis has the influence on the results themselves. We find that there are four aspects causing
the difference in our prediction results when we check the process and code we used: (1) the environmental covariables applied
for each prediction model; (2) the predicted ILR components of the testing sets; (3) the back-transformed values for the three
components of soil PSFs; and (4) the predicted ILR residuals (testing sets) without back transformation (only for the RK
method).
For (1). The three ILR balances generated different transformed datasets. The GLM model used the "glmStepAIC" algorithm
(i.e., a stepwise regression) to select the best combination of environmental covariables for each ILR component. (**P8L325**"*The
Akaike's information criterion (AIC) was applied to choose the best predictors and remove model multicollinearity using a
backward stepwise algorithm."*) Therefore, the variable inputs were different for these ILR data. We listed the choice of
variables of each ILR for one random prediction (Table 1).
**Table 1.** Combination of environmental covariables for different ILR data.

| Data | Combination of environmental covariables |
|------|------------------------------------------|
| ILR1SBP1 | WWC + ndvi + lon + soc + rain + CNB + NH |
| ILR2SBP1 | FWHC + WWC + ndvi + tem + soc + dem + rain + AHS + aspect + MSP |
| ILR1SBP2 | FWHC + WWC + ndvi + tem + soc + SHC + dem + rain + AHS + aspect + MSP |

| ILR2SBP2 | FWHC + WWC + lon + soc + aspect + CNB + MSP + MRVBF |
|---|---|
| ILR1SBP3 | FWHC + WWC + tem + lat + soc + dem + aspect + CNB + MSP + MRVBF |
| ILR2SBP3 | ndvi + tem + soc + SHC + dem + rain + aspect + MSP + SH |

For (2) and (3). Moreover, an independent dataset validation was used for the accuracy assessment in this study. The training and testing sets were entirely different and had no intersection. Therefore, the predicted ILRs in the testing sets were different and the back-transformed soil PSFs and the accuracy indicators (ME and RMSE) were also different (see response 1).

For (4). For the validation and prediction maps of RK, the results were the sum of the predicted ILR and ILR residuals, which were then back-transformed, producing different values (Fig. 2). We also noticed that although the differences among the values were small, the inverse transformation can enlarge the difference and prediction errors because of the value ranges.

[Figure]

**Fig. 2.** Process of RK method in our study.

In summary, we think the reasons for the different results start with the first step (EC selection), and affect the next steps. We have added more explanation for this in our revised version.

**P20L514:** *"The results of GLM and GLMRK should not depend on the ILR basis being chosen, which has been proved by*

*previous studies on the use of linear models and kriging for compositional data (Pawlowsky-Glahn et al, 2015). However, the*

*GLM model used the "glmStepAIC" algorithm (i.e., a stepwise regression) to select the best combination of environmental*

*covariables for each ILR component. Therefore, the variable inputs were different for these ILR data, and further impact the*

*accuracy assessment and prediction maps."*

**Reference**

Pawlowsky-Glahn V, Egozcue JJ, Tolosana-Delgado R.: Modeling and analysis of compositional data. John Wiley & Sons,

Ltd, 2015.

**Comment 3.**

**(1)** Studying the bias of linear models and regression kriging is not meaningful, because both are unbiased methods. If bias is found, it derives from an incorrect definition of the notion of bias, which should be considered in the geometry of the simplex.

**(2)** Moreover, all the statistics and summaries should be considered in a multivariate setting, and the consideration of univariate components of psfs should be completely avoided (particularly if the aim is to approach them in a compositional setting).

Overall, the paper does not discuss clearly the background on compositional data analysis, and the comments to analyses and results are often formally inappropriate, showing inconsistencies and general confusion on the concepts related with the theory of compositional data analysis.

**Response:** (1) We agree that the linear models and RK are unbiased. However, in the validation method used in this study, an independent dataset validation was used for the accuracy assessment. Therefore, the training (70%) and test (30%) sets were entirely different and had no intersection. Although these models are unbiased, we can also verify the bias of an independent dataset (predictions) using the mean error (ME). In other words, for spatial interpolation, the usual methods of validation for comparing the interpolation methods are known as cross-validation and validation with an independent data set. Cross- validation involves eliminating each observation in turn, estimating the value at its site from the remaining observations and comparing the predicted value with the measured value. This procedure is a rapid, inexpensive one for comparing predicted and measured values. Unfortunately, it has limitations in many cases. For kriging estimators, it retains the same variogram, and to be true cross-validation the variogram should be recomputed and fitted afresh when each observation is removed. These shortcomings can be avoided by using an independent data set for validation. Validation with an independent data set which is a superior and more dependable method directly estimates the spatial uncertainty, as validation points are located randomly throughout the field (Shi et al., 2009). Therefore, the concept of unbiased is for all sampling points, not for the validation.

(2) Furthermore, for the statistics and summaries, the Aitchison Distance (AD) was applied as an indicator to evaluate the overall performance of the models. The AD can consider a multivariate setting. In addition, we also wanted to evaluate and compare which component (i.e., sand, silt, and clay) performed best among these prediction models. In the field of soil PSF

spatial prediction, each component should be evaluated and not just the overall impact, which will help to fully understand the modeling process. The three ILR balances produced different ILR data, with distinct data ranges and other statistical characteristics. This is why we explored whether different balances would affect one soil PSF component and further improve the accuracy.

We have listed some previous studies that used ME to evaluate soil PSF prediction bias for a linear regression (LR) method combined with a log-ratio, which confirms that the use of these univariate metrics should not be avoided (Buchanan et al.,

2012; Huang et al., 2014).

**Refrence**

Buchanan, S., Triantafilis, J., Odeh, I. O. A., and Subansinghe, R.: Digital soil mapping of compositional particle-size fractions using proximal and remotely sensed ancillary data, Geophysics, 77, WB201-WB211, 10.1190/geo2012-0053.1, 2012.

Huang, J., Subasinghe, R., and Triantafilis, J.: Mapping Particle-Size Fractions as a Composition Using Additive Log-Ratio

Transformation and Ancillary Data, Soil Sci. Soc. Am. J., 78, 1967-1976, 10.2136/sssaj2014.05.0215, 2014.

Shi, W., Liu, J., Du, Z., Song, Y., Chen, C., and Yue, T.: Surface modelling of soil pH, Geoderma, 150, 113-119,

10.1016/j.geoderma.2009.01.020, 2009.

**Comment 4.** The discussion is very confused, and the overall message strongly hindered by incorrect English wording.

**Response:** Thanks for the suggestion about the quality of the English language of this paper. We looked for some senior editors from a professional English polishing company to improve the overall language of this article and we have checked and improved the writing in the revised version.

[Figure]

**EDITORIAL CERTIFICATE**

This document certifies that the manuscript below was edited for correct English language usage, grammar, punctuation and spelling by qualified native English speaking editors at Charlesworth Author Services.

**Paper Title:**

Compositional balance should be considered in soil particle-size fractions mapping using hybrid interpolators

Author:
Wenjiao Shi

**Date certificate issued:**

Nov. 23, 2020

[cwauthors.com](cwauthors.com)

Special thanks to you for your kind comments.

Yours sincerely,

Wenjiao Shi

E-mail: shiwj@lreis.ac.cn

**Compositional balance should be considered in the mapping of soil particle-size fractions  using hybrid interpolators**

Mo Zhang[1,2], Wenjiao Shi[1,3]

[1]Key Laboratory of Land Surface Pattern and Simulation, Institute of Geographic Sciences and Natural Resources Research, Chinese Academy of Sciences, Beijing 100101, China
[2]School of Earth Sciences and Resources, China University of Geosciences, Beijing 100083, China
[3]College of Resources and Environment, University of Chinese Academy of Sciences, Beijing 100049, China

*Correspondence to:* Wenjiao Shi (shiwj@lreis.ac.cn), Institute of Geographic Sciences and Natural Resources Research, Chinese Academy of Sciences. 11A, Datun Road, Chaoyang District, Beijing 100101, China.

**Abstract**. Digital soil mapping of soil particle-size fractions (PSFs) using log-ratio methods is a widely used technique. As a hybrid interpolator, regression kriging (RK) is an alternative way to improve prediction accuracy. However, there is still a lack of systematic comparisons and recommendations when RK is applied for compositional data, and it is not known if the performance based on different balances of isometric log-ratio (ILR) transformation is robust. Here, we systematically compared the generalized linear model (GLM), random forest (RF), and their hybrid pattern (RK) using different balances of ILR transformed data for soil PSFs, with 29 environmental covariables (ECs) for the prediction of soil PSFs in the upper reaches of the Heihe River Basin. The results showed that RF performed best, with more accurate predictions, but GLM produced a more unbiased prediction. For the hybrid interpolators, RK was recommended because it widened the data ranges of the prediction results, and modified the bias and accuracy for most models, especially for RF. However, there was a drawback due to the data distributions and model algorithms. Moreover, prediction maps generated from RK revealed more details of the soil sampling points. For  three components, sequential binary partition (SBP) based ILR transformed data produced different distributions, and it is not recommended to use the most abundant component of compositional data as the first component of a permutation. This study provides a reference for the spatial simulation of soil PSFs combined with ECs and transformed data at the regional scale.

**1 Introduction**

Recently, spatial interpolation of soil particle-size fractions (PSFs) has become a focus of  soil science researchers. More accurately predicted soil PSFs could contribute to a better understanding of hydrological, physical, and environmental processes (Delbari et al., 2011; Ließ et al., 2012; McBratney et al., 2002).

The characteristics of compositional data makes soil PSFs  more impressive than other soil properties.

Soil PSFs are usually expressed as three components of discrete data – sand, silt, and clay, and carry only relevant percentage information. Soil texture is classified as soil PSFs, which can be demonstrated on a ternary diagram. The closure system of a ternary diagram is not Euclidean space, but is rather Aitchison space (i.e., the simplex) (Aitchison, 1986). Due to "spurious correlations" (Pawlowsky-Glahn, 1984), traditional statistical methods based on Euclidean geometry may generate mistakes when dealing directly with soil PSF data (Filzmoser et al., 2009). The requirement for constant sum, nonnegative, unbiased values is the key to spatial interpolation (Walvoort and de Gruijter, 2001). Data transformation is crucial for the transformation of compositional data from the simplex to the real space. Log ratio transformations play a significant role in compositional data analysis, including the additive log-ratio (ALR), centered log-ratio (CLR) (Aitchison, 1986), and isometric log-ratio (ILR) (Egozcue et al., 2003).

Although these three log-ratio methods have been widely applied to transform soil PSF data, different study area scales and model selection should be considered when modeling. For local-scale study areas, geostatistical models, i.e., ordinary kriging (OK) and compositional kriging, combined with log-ratio transformed data, are sufficient to map spatial patterns virtually, as shown in our previous study (Wang and Shi, 2017). As another perspective, functional compositions combined with the kriging method can also be applied to produce soil particle size curves (PSCs) (Menafoglio et al., 2014), providing an abundance of information. This involves the use of complete and continuous information rather than discrete information, and soil PSFs can be extracted from the predicted soil PSCs (Menafoglio et al., 2016a). Log-ratio transformations can also be combined with functional-compositional data for the stochastic simulation of PSCs (Menafoglio et al., 2016b, Talska et al., 2018). For middle-scale study areas, outliers may lead to the overestimation of the variogram, resulting in prediction errors (Lark, 2000). Therefore, the spatial interpolation should take robust variogram estimators into account to improve model performance (Lark, 2003). A previous study proved that applying robust variogram estimators in log-ratio co-kriging significantly improved mapping performance (Wang and Shi, 2018). For large-scale study areas, geostatistical models are limited by the number of soil sampling points and increased spatial variability. An increasing number of studies have concentrated on mapping soil PSFs using different machine learning models, statistical models, and geostatistical models combined with ancillary data (i.e., environmental covariables, ECs) on a broad basin scale (Zhang et al., 2020), national scale (Akpa et al., 2014), and global level (Hengl et al., 2017) using log-ratio transformed data.

Among these EC-combined models, linear, machine-learning, geostatistical models, and high accuracy surface modeling (Yue et al., 2020) have been commonly used in middle or large-scale studies. Linear models, such as the generalized linear model (GLM) and multiple linear regression (MLR) have been used in soil PSF predictions because of their flexibility and interpretability (Lane, 2002; Buchanan et al., 2012). Many machine-learning models have been applied for the interpolation of soil PSFs and soil texture classification. For example, tree learners, such as the random forest (RF), have been shown to be advantageous due to their ability to handle noisy datasets and generate more realistic maps (Zhang et al., 2020). Furthermore, regression kriging (RK)
can not only combine ECs through its regression function, but it also improves model
accuracy as a hybrid interpolator for some soil properties such as topsoil thickness and pH (Hengl et al., 2004). However, the
scope of the comparison needs to be expanded to further explore the accuracy and predict
compositional data using linear models, machine-learning models, and other models combining RK (hybrid
patterns).

In log-ratio methods, the ILR method performs better than ALR and CLR in both  theory and in practice
(Filzmoser and Hron, 2009; Wang and Shi, 2018; Zhang et al., 2020). The ILR method eliminates model collinearity and
preserves advantageous properties such as isometry, scale invariance, and sub-compositional coherence,
through its use of orthonormal coordinate systems (i.e., balances) using a sequential binary partition (SBP)
(Egozcue and Pawlowsky-Glahn, 2005). These choices are not unique. In other words, multiple sets of ILR transformed data
can be generated by permutations of components (different SBPs) in the compositional data. The choice of an
SBP can be based on prior expert knowledge, using a compositional biplot (Lloyd et al., 2012) or variograms and cross-
variograms (Molayemat et al., 2018). It has been proven in statistical science that different results are obtained using
different choices of SBP balances, and the option of a specific SBP for data compositions is crucial for the intended
interpretation of coordinates (Fiserova and Hron, 2011). However, most  soil science researchers have ignored
this point. Martins et al. (2016) reported that  clay has been widely used as the denominator in the ALR method
because it is typically the most abundant component of compositions. Few studies have compared the different SBP options
from the perspective of accurate assessments and analyzed whether these differences are due to the general
characteristics of specific data sets or log-ratio transformations.

Therefore, based on our previous work, the objectives of this study were to: (i) compare the spatial prediction
accuracy of soil PSFs using a GLM and RF combined with
ECs and ILR transformed data; (ii) determine whether hybrid interpolators (GLMRK and RFRK) can improve the
interpolation performance of a GLM and RF; and (iii) explore the distributions of different transformed data and the variation
law of precision based on different choices of SBP balances of ILR.

**2 Methods and materials**

**2.1 Study area**

The study area was the upper reaches of the Heihe River Basin (HRB), which is the source of the Heihe River
and the central area of  runoff generation in the HRB. The elevation in this area ranges from 1640  to 5573 m (Fig. 1),
and the climate is damp and cold being dominated by the Qilian Mountains. The mean annual rainfall in the study area
is 350 mm, and the mean annual temperature is lower than 4-°C. Meadow and steppe are the dominant vegetation
types. Grassland is the primary  land use. The main soil classes are frigid calcic soil in the southwest of the study area, with cold desert soil dominatesdominating the southeast, andwhile Castanozems and Sierozems mainly distributeare
distributed in the north of the study area.

[Figure]

**Figure. 1.** The location, elevation, and soil samples on the upper reaches of the Heihe River Basin.
**2.2 Data collection and analysis**

**2.2.1 Soil PSF data**

A total of 262 soil samples based on a purposive sampling strategy were collected in the upper reaches of the HRB based on a
purposive sampling strategy and were used to characterize the spatial variability of soil PSFs at the regional- scale study area
(Fig. 1). The variability of soil formation factors, such as the elevation, soil classestype, vegetation classesclass, and
geomorphology classes of the upper reaches of the HRB was considered in soil samplessample collection. The average of three
mixed three topsoil samples (approximately approximate depth of 0—20 cm) was obtained to reduce the noise of soil
samplessample parameters, and thea parallel sample was also measured. Subsequently, about 30 g of each soil sample was air-
dried, and the chemical and physical analyses were operated after the fieldwork. Collected conducted in the laboratory. Soil
PSF information was obtained for the soil samples recorded the information about soil PSFs using a Malvern Panalytical
Mastersizer 2000 laser. with less than 3-% average measurement error.

**2.2.2 The selection of ECs**

There were 29 ECs considered in our study, including both continuous and categorical variables (Table 1). They followed the principles of the SCORPAN model (McBratney et al., 2003), which  is defined as $S_a = f(S, C, O, R, P, A, N)$. $S_a$ are soil attributes (or classes) as a function of soil properties ($S$) or other properties, i.e., climatic properties ($C$), organisms and vegetation ($O$), relief such as topography and landscape attributes ($R$), parent material ($P$), an age or time factor ($A$), and spatial position ($N$). The continuous variables included the morphometry and hydrologic characteristics of topographic properties, climatic and vegetative indices, and soil physical and chemical properties. The categorical variables included geomorphology, land use types, and vegetation classes, which were transformed from vector to raster (1000 m). Due to the intricate patterns of topography in the upper reaches of the HRB, the variable of topographic properties dominated the ECs. The System for Automated Geoscientific Analyses geographic information system (SAGA GIS) (Conrad et al., 2015) was applied for a terrain analysis to derive topographic variables using the 30 m resolution Advanced Spaceborne Thermal Emission and Reflection Radiometer Global Digital Elevation Model (ASTER GDEM, http://www.gscloud.cn). A collinearity test removed the redundant variables, and the topographic properties were then resampled to 1000 m. More details of the ECs are provided in the Data Availability section.

**Table 1.** Selected environmental covariates in our study.

| Representation | Environment covariables | Abbreviation |
| --- | --- | --- |
| Morphometry characteristics | Analytical Hill Shading | AHS |
| | Aspect | ASPECT |
| | Closed Depressions | CD |
| | Convergence Index | CI |
| | Channel Network Base Level | CNB |
| | Slope Length and Steepness Factor | LSF |
| | Multi-resolution Ridge Top Flatness Index (Gallant and Dowling, 2003) | MRRTF |
| | Multi-resolution Valley Bottom Flatness Index (Gallant and Dowling, 2003) | MRVBF |
| | Mid-slope Position | MSP |
| | Plan Curvature | PLC |
| | Profile Curvature | PRC |
| | Slope Height | SH |
| | Slope Length (D. Moore et al., 1993) | SL |
| | Tangential Curvature (Florinsky, 1998) | TC |
| Hydrologic characteristics | Catchment Area | CA |
| | Surface Area | SA |

| | Stream Power Index | SPI |
| --- | --- | --- |
| | Topographic Wetness Index (Beven and Kirkby, 1979) | TWI |
| | Vertical Distance to Channel Network | VDCN |
| Climatic and vegetative indices | Average Annual Precipitation | RAIN |
| | Average Annual Temperature | TEM |
| | Normalized Differential Vegetation Index | NDVI |
| Soil physical and chemical properties | Field Water Holding Capacity (Yi et al., 2015; Song et al., 2016; Yang et al., 2016) | FWHC |
| | Soil Depth (Yi et al., 2015; Song et al., 2016; Yang et al., 2016) | PDEPTH |
| | Saturated Hydraulic Conductivity (Yi et al., 2015; Song et al., 2016; Yang et al., 2016) | SHC |
| | Soil Organic Carbon | SOC |
| Categorical maps | Geomorphology | GEOT |
| | Land Use | LU |
| | Vegetation Classes | VEGET |

**2.3 Isometric log-ratio transformation and SBP**

An orthonormal basis of the ILR was chosen to isometrically project the compositions from $S^D$ (the simplex for the Aitchison geometry) to $R^{D-1}$ (real space for the Euclidean geometry). The choice of a specific orthonormal basis for use on $S^D$ can be explained by the SBP for the groups of compositions (Egozcue and Pawlowsky-Glahn, 2005). The  choice of the construction of coordinates (i.e., balances) between groups of compositions was calculated as follows:

$$z_k = \sqrt{\frac{r_k s_k}{r_k + s_k}} \, ln\left(\frac{(x_{i_1} x_{i_2} \dots x_{i_{r_k}})^{1/r_k}}{(x_{j_1} x_{j_2} \dots x_{j_{s_k}})^{1/s_k}}\right), \; k = 1, \dots, D-1, \tag{1}$$

where $z_k$ refers to the balance between two groups; $i_1, i_2, \dots, i_{r_k}$ is the $r_k$ part of one group; and $j_1, j_2, \dots, j_{r_k}$ is the $s_k$ part of the other group. Therefore, in a stepwise manner, the balances contain  all the relevant information of the compositions in two groups. This can also  be explained in a tabular form. For soil PSF data (D = 3), all three choices of the balance of SBPs are shown in Table 2. The first component of the ILR contained all the information on soil PSFs, and the main difference in the choice of balances for soil PSFs was the order of the three parts, i.e., the first order of the soil PSF component was used as the numerator of the first ILR equation. In our study, three SBP balances —, SBP1, SBP2, and SBP3, were transformed from the original soil PSF data, and the orders of soil PSF data were $(sand, silt, clay)$, $(silt, clay, sand)$, and $(clay, sand, silt)$, respectively. The transformation equations for the ILR

can be derived from Eq. (1), and were defined as Eqs. (2) and  (3). The inverse equations for ILR were defined as Eqs. (4), (5), (6). The ILR transformation and its inverse were conducted using the R package

"compositions" (K. Gerald van den Boogaart and Raimon Tolosana, 2014).

$\mathbf{z} = (z_1, \ldots z_{D-1}) = ILR(\mathbf{x})$, and for $i = 1, \ldots, D-1$ and component $x_i$, (2)

$$z_i = \sqrt{\frac{D-i}{D-i+1}} \, ln \frac{x_i}{\sqrt[D-i]{\Pi_{j=i+1}^{D} x_j}}.$$ (3)

$$Y(x_j) = \sum_{j=1}^{D} \frac{ILR(x_j)}{\sqrt{j \times (j+1)}} - \sqrt{\frac{j-1}{j}} \times ILR(x_j),$$ (4)

$$ILR(x_0) = ILR(x_D) = 0,$$ (5)

$$\overline{ILR}(x_j) = \frac{exp(Y(x_j))}{\Sigma_{j=1}^{D} exp(Y(x_j))}.$$ (6)

**Table 2** All choices of SBPs for soil PSF data (D = 3), the orders of soil PSFs data are $(sand, silt, clay)$, $(silt, clay, sand)$

and $(clay, sand, silt)$ for SBP1, SBP2 and SBP3.

| Groups | Step | Sand | Silt | Clay | r | s | Balance |
|--------|------|------|------|------|---|---|---------|
| SBP1 | 1 | + | - | - | 1 | 2 | Step1: $z_1 = \sqrt{\frac{2}{3}} \, ln \frac{sand}{\sqrt{silt \times clay}}$ |
|  | 2 | 0 | + | - | 1 | 1 | Step2: $z_2 = \sqrt{\frac{1}{2}} \, ln \frac{silt}{clay}$ |
| SBP2 | 1 | - | + | - | 1 | 2 | Step1: $z_1 = \sqrt{\frac{2}{3}} \, ln \frac{silt}{\sqrt{clay \times sand}}$ |
|  | 2 | - | 0 | + | 1 | 1 | Step2: $z_2 = \sqrt{\frac{1}{2}} \, ln \frac{clay}{sand}$ |
| SBP3 | 1 | - | - | + | 1 | 2 | Step1: $z_1 = \sqrt{\frac{2}{3}} \, ln \frac{clay}{\sqrt{sand \times silt}}$ |
|  | 2 | + | - | 0 | 1 | 1 | Step2: $z_2 = \sqrt{\frac{1}{2}} \, ln \frac{sand}{silt}$ |

**2.4 Linear model, machine-learning model, and hybrid patterns**

**2.4.1 Generalized linear model**

The GLM is an extended version of the linear model, which contains response variables, with non- normal distributions (Nelder and Wedderburn, 1972). The link function is embedded into the GLM to ensure the classical linear model assumptions. The scaled dependent variables and the independent variables can be connected using a link function for the additive combination of model effects, the choice of link function depends on the distribution of response variables (Venables and Dichmont, 2004). A Gaussian distribution with an identity link function was applied in our study, which produced consequences equivalent to that of MLR (Nickel et al., 2014). However, categorical variables can be directly trained in the GLM without setting dummy variables. The Akaike's information criterion (AIC) was applied to choose the best predictors and remove model multicollinearity using a backward stepwise algorithm.

**2.4.2 Random forest**

The RF is a non-parametric technique, which combines the bagging method with a selection of random variables as an extended version of a regression tree (RT) (Breiman, 1996, 2001). It can improve model prediction accuracy by producing and aggregating multiple tree models. The principle of the RF is to merge a group of "weak trees"

together to generate a "powerful forest." The bootstrap sampling method was applied for each tree, and each predictor was selected randomly from all model predictors. The "out of bag" (OOB) data were applied to produce reliable estimates in an internal validation using a random subset independent of the training tree data. Three parameters needed to be tuned:  number of trees ($ntree$); minimum size of terminal nodes ($nodesize$), and  number of variables randomly sampled as predictors for each tree ($mtry$) (Liaw and Wiener, 2001). The standard value of the $mtry$ parameter was one-third of the total number of predictors, while $ntree$ and $nodesize$ were 500 and 5, respectively. For regression, the mean square errors (MSEs) of predictions were estimated to train the trees. The variable importance of the RF

was produced from the OOB data using the "importance" function. One of the benefits of the RF is that the ensembles of trees are used without pruning to ensure that the most significant amount of variance can be expressed. Moreover, the RF can reduce model overfitting and normalization is unnecessary due to the  effects on the value range being insensitive. The GLM and RF algorithms  and the parameter adjustment of the RF were conducted in the R package "caret" (Max Kuhn, 2018).

**2.4.3 Regression kriging**

Regression kriging  is a hybrid interpolation technique that combines regression models (e.g., GLM and RF) with the OK of the residuals of regression models (Odeh et al., 1995). Mathematically, the RK method corresponds to two interpolators, the regression part and the kriging part, which are operated separately (Goovaerts, 1999). One limitation of using only the regression part is that it is usually only useful within the range of values of the training sets (Hengl et al., 2015). The principle of the RK method is that the regression model explains a deterministic component of spatial variability, and the interpolation of regression residuals generated from OK is used to describe the spatial variability (Bishop and

McBratney, 2001; Hengl et al., 2004). The residuals are used to create a variogram (e.g., Gaussian, spherical, or exponential) for models based on the MSE from the results of a cross-validation. First, the regression part in our study (GLM or RF) was used to predict soil PSFs. The residual from the fitted model was then calculated by subtracting the regression part from the observations. Subsequently,  OK was applied for the whole study area to interpolate the residuals. Finally, the regression prediction and the predicted residuals at the same location were summed.

The variograms of the RK method were generated automatically  using the "autofitVariogram" function in the R package
"automap" (Hiemstra et al., 2009).

**2.5 Prediction method system and validation**

The method system of spatial interpolation models for soil PSFs is presented in Table 3. We systematically
compared 12 models: four interpolators, including GLM and RF  with or without RK, and three SBPs of the ILR
transformation method. For the validation of model performance, the independent data set validation was used to evaluate the
prediction bias and accuracy of the models. The sub-training sets (70%) and the sub-testing sets (30%) were randomly
selected from data independently, and this process was repeated 30 times.

**Table 3.** The method system of spatial interpolation models of soil PSFs.

| Models | GLM | GLMRK | RF | RFRK |
|---|---|---|---|---|
| ILR_SBP1 | GLM_SBP1 | GLMRK_SBP1 | RF_SBP1 | RFRK_SBP1 |
| ILR_SBP2 | GLM_SBP2 | GLMRK_SBP2 | RF_SBP2 | RFRK_SBP2 |
| ILR_SBP3 | GLM_SBP3 | GLMRK_SBP3 | RF_SBP3 | RFRK_SBP3 |

The mean error (ME), the root mean square error (RMSE), and Aitchison distance (AD) were used to evaluate and compare
the prediction performance of models. The ME and RMSE measure prediction bias and accuracy, respectively (Odeh et al.,
1995). The AD is an overall indicator of compositional analysis, which describes the distance between two data compositions.
Generally, in an accurate, unbiased model  all three values will be close to 0. The  ME, RMSE,
and AD were calculated as follows:

$$ME = \frac{1}{n}\sum_{i=1}^{n}(M_i - P_i), \tag{7}$$

$$RMSE = \sqrt{\frac{1}{n}\sum_{i=1}^{n}(M_i - P_i)^2}, \tag{8}$$

$$AD = \left[\sum_{i=1}^{D}(log\frac{M_i}{G(\boldsymbol{M})} - log\frac{P_i}{G(\boldsymbol{P})})^2\right]^{0.5}, \tag{9}$$

where $M_i$ and $P_i$ are the measured  and predicted values at the $i$th position, respectively; $n$ refers to the number
of soil samples; $D$ is the number of dimensions of data compositions; and $G(\boldsymbol{M})$ and $G(\boldsymbol{P})$ denote the geometric
mean with the form $G(\mathbf{x}) = $ $(x_1,\ldots,x_D)^{1/D}$ of the measured and predicted values, respectively.

**2.6 Statistical analysis**

The interpretation of the balances of ILR is based on a decomposition of the covariance (COV) structure (Fiserova and
Hron, 2011). We calculated the variance (VAR),  (COV), and the corresponding correlation coefficient (CC)
of ILR transformed data based on different SBP balances . The equations for calculating VAR, COV, and CC

were derived from Eq. (1) as follows:

$VAR(z) = \frac{1}{r+s}\sum_{p=1}^{r}\sum_{q=1}^{s} var(ln\frac{x_{ip}}{x_{jq}}) - \frac{s}{2r(r+s)}\sum_{p=1}^{r}\sum_{q=1}^{r} var(ln\frac{x_{ip}}{x_{iq}}) - \frac{r}{2s(r+s)}\sum_{p=1}^{s}\sum_{q=1}^{s} var(ln\frac{x_{jp}}{x_{jq}}) -$

$\frac{r}{2s(r+s)}\sum_{p=1}^{s}\sum_{q=1}^{s} var(ln\frac{x_{jp}}{x_{jq}})$      (10)

$COV(z_1, z_2) = \frac{C}{2r_1 s_2}\sum_{p=1}^{r_1}\sum_{q=1}^{s_2} var(ln\frac{x_{ip}^1}{x_{jq}^2}) + \frac{C}{2r_2 s_1}\sum_{p=1}^{r_2}\sum_{q=1}^{s_1} var(ln\frac{x_{ip}^2}{x_{jq}^1}) - \frac{C}{2r_1 r_2}\sum_{p=1}^{r_1}\sum_{q=1}^{r_2} var(ln\frac{x_{ip}^1}{x_{iq}^2}) -$

$\frac{C}{2s_1 s_2}\sum_{p=1}^{s_1}\sum_{q=1}^{s_2} var(ln\frac{x_{jp}^1}{x_{jq}^2}),$      (11)

$CC = \frac{COV(z_1, z_2)}{\sqrt{var(z_1) \cdot var(z_2)}}$      (12)

For soil PSF data, Eqs. (10), (11), and (12) can be simplified to three dimensions. The relationship between the ratios of soil PSF components and the dominant roles of ILR transformed data were indicated from the covariance structure. All the statistical analyses, such as the descriptive statistics of soil PSF data, calculation and evaluation of indicators, and the spatial operation of prediction maps, were performed using the R statistical program (R

Development Core Team, 2019).

**3 Results**

**3.1 Exploratory data analysis**

**3.1.1 Descriptive statistics of soil PSF data**

From the descriptive statistics of the original (raw) and ILR transformed data, the silt fraction  dominated the soil

PSFs , accounting for a more substantial amount than the sand and clay fractions. The distributions of the sand and clay fractions were similar (Fig. 2a). The ILR transformed data based on the three SBP balances revealed different distributions (Figs. 2b, 2c, and 2d). For example, two ILR components  (ILR1 and ILR2) for SBP1

had a symmetric distribution around zero  at the $x$-axis (Fig. 2b). In comparison, the distribution of data generated from

SBP2 or SBP3 had a mirrored symmetry, with a left-skewed ILR1 of SBP2 and right-skewed

ILR2 of SBP3 (Figs. 2c and 2d). The comparison of means and medians demonstrated that the back-transformed means of three sets of ILR transformed data were the same, and the mean ILR of sand  was closer to the median compared with the original soil PSF  data. In contrast, the opposite patterns were apparent for the silt and clay components (Fig. 2e).

[Figure]

**Figure. 2.** Descriptive statistics of original soil PSF data and ILR transformed data using different balances of SBP**.** Not that means of Sand_ILR, Silt_ILR, and Clay_ILR from different SBPs of ILR were back-transformed to the real space.

**3.1.2 Covariance structure of ILR transformed data with different balances**

The covariance analysis of the transformed data of soil PSFs  based on the different SBPs showed that the variance VarILR_1 of SBP3 was the largest, followed by the  VarILR_1 of SBP1 and SBP2 (Table 4). The variance of the second component of ILR (VarILR_2) followed the opposite pattern to that of VarILR_1. The   and  corresponding  (CC) followed the same pattern of SBP1 > SBP3 > SBP2. From these values, the relationships among soil PSF components or ratios were revealed. The first ILR equation  ($z_1$ in Table 2) contained all the soil PSF information , while the second one ($z_2$ in Table 2) included only two components. The VarILR_1 information was therefore more abundant. Six values of VarILR_1 and VarILR_2 were not 0 (or not nearly 0), indicating that there was no constant (or almost  constant) value in any two ratios of soil PSF components. The COV value of  SBP3 was close to 0, indicating that the proportions of *clay/sand* and *clay/silt* were approximately the same. The same results were generated from the corresponding CC.

**Table 4** Covariance analysis of soil PSF data based on different SBPs. VarILR_1 and VarILR_2 denote the variance of the first and the second component of ILR, respectively. COV refers to the covariance of ILR1 and ILR2. CC is the correlation coefficient.

| Balances | VarILR_1 | VarILR_2 | COV | CC |
|---|---|---|---|---|
| SBP1 | 0.53 | 0.71 | 0.32 | 0.52 |

| | | | | |
|---|---|---|---|---|
| SBP2 | 0.39 | 0.86 | -0.24 | -0.41 |
| SBP3 | 0.94 | 0.30 | -0.09 | -0.16 |

The distribution of soil PSF sampling data in a ternary diagram (the United States Department of Agriculture (USDA) texture triangle) showed that the main texture class was silt loam (Fig. 3a). The biplot of soil samples demonstrated that the rays of the three components, i.e., sand, silt, and clay, were reasonably well clustered at about 120-° in the three groups (Fig. 3b).

[Figure]

**Figure. 3.** The distribution in the USDA triangle (a) and biplot graph (b) of soil PSFs sampling. The red, smooth curve of these soil samples in the USDA triangle was fitted by loess function in R.

**3.2 Accuracy comparison of different models using ILR data**

The first three rows of the boxplots  Figs. 4a, 4b, and 4c indicate the bias of the different models according to their ME values. The ME of sand was closest to 0, followed by the MEs of clay and silt. The GLM was more unbiased than the RF, with lower ME values. After combining with RK, there was an improvement in the ME for  most GLM and RF models (Figs. 4a, 4b, and 4c). For the accuracy assessment, the RMSE of silt was higher than for the other two components. The GLMRK did not perform as well as  expected in terms of the RMSE, with only  the sand component having an improved RMSE (Fig. 4d). However, the RFRK performed better than the GLMRK and improved the RMSE of most  parts compared with the RF, except for the RFRK_SBP1 of sand. As an overall indicator of soil PSFs, the AD showed that the RF (or RFRK) performed better than the GLM (or GLMRK) in terms of both average RMSE values and uncertainties (Fig. 4g). Moreover, the RFRK improved the AD values for the SBP2 and SBP3 methods. For the uncertainty assessment, the RF generated fewer difficulties than the GLM, and the models combined with RK further reduced the uncertainties for most GLM and RF models.  The model performances were different for the three SBP

balances. To better evaluate model performance using the different SBP balances, we graded each box from 1 to 3, and the
final results wereare shown in the Supplementary Material table(Table S1.1. It). The results demonstrated that SBP1 performed
best, with the lowest ME value amongof all models. For the accuracy comparison, the pattern is not there was no apparent
pattern, but it canaccuracy could be considered hierarchically. For the GLM, SBP1 hadperformed better performance than the
other two SBPsSBP methods, which also performed well when RK was added (GLMRK). For RF, SBP1 produced the best
result. However, the introduction of RK maderesulted in SBP3 performedperforming best among the three methods. Further,
theThe RMSEs generated from RFRK using SBP3 data had the best accuracy among all the models in our study.

[Figure]

**Figure. 4.** Accuracy comparison of GLM, RF, and their RK patterns using different ILR balances data. The mean values of
different model indicators were calculated in their boxes.

**3.3 Spatial prediction maps of soil PSFs generated from the different models**

Prediction maps of soil PSFs made from the different models are shown in Figs. 5, 6, and 7. For the components of soil PSFs, the maps of the three group maps followed a similar rule. The GLM and GLMRK produced more extensive ranges of predicted values, and their maps were more relevant to the real environment. However, the RF and RFRK predicted a relatively narrow range of low values for these components, revealing a smoother distribution than   that generated by the GLM and GLMRK. Unlike the regression methods , the RF and RFRK methods produced hot and cold spots on the prediction maps and more details of the soil sampling points were apparent (Fig. S2.1) .

[Figure]

**(a) GLM_SBP1**  **(b) GLM_SBP2**  **(c) GLM_SBP3**

**(d) GLMRK_SBP1**  **(e) GLMRK_SBP2**  **(f) GLMRK_SBP3**

**(g) RF_SBP1**  **(h) RF_SBP2**  **(i) RF_SBP3**

**(j) RFRK_SBP1**  **(k) RFRK_SBP2**  **(l) RFRK_SBP3**

**Sand (%)**

| | |
|---|---|
| 0 - 11 | 16 - 20 | 24 - 29 | 36 - 50 |
| 11 - 16 | 20 - 24 | 29 - 36 | 50 - 75 |

40  80  160  240  320  km

**Figure. 5.** Spatial prediction maps of the sand component of the upper reaches of the Heihe River Basin.

[Figure]

**Figure. 6.** Spatial prediction maps of the silt component of the upper reaches of the Heihe River Basin.

[Figure]

| Clay (%) | | | | |
|---|---|---|---|---|
| 0 - 5 | 8 - 11 | 14 - 18 | 21 - 26 | 31 - 41 |
| 5 - 8 | 11 - 14 | 18 - 21 | 26 - 31 | 40 - 60 |

40   80      160      240      320
km

**Figure. 7.** Spatial prediction maps of the clay component of the upper reaches of the Heihe River Basin.

**3.4 Spatial distribution of soil texture classes in the USDA triangles**

The predicted soil textures based on the USDA texture triangles (Fig. 8) showed that most predictions fell within the range of observed soil textures (Fig. 3a), and silt loam was the dominant soil texture in all cases. The GLM produced a more discrete distribution than the RF, and the RK method expanded the effect of dispersion. In the trends of the predicted samples, the silt components predicted from all models were overestimated. The pattern fitting curves indicated that the prediction results were closer to the bottom right of the

USDA soil texture triangle than the soil PSF observations. The GLMRK and RFRK curves were longer than the GLM and RF curves, with a more extensive range of values in the ternary diagram. Compared with the GLMRK, the RFRK produced a more upward extension (Figs. 8j, k, l). It was clear that the clay fraction was overestimated and the sand fraction was underestimated.

[Figure]

**Figure. 8.** Predicted 262 soil samples based on leave-one-out method in USDA texture triangles using (a) GLM_SBP1, (b) GLM_SBP2, (c) GLM_SBP3, (d) GLMRK_SBP1, (e) GLMRK_SBP2, (f) GLMRK_SBP3, (g) RF_SBP1, (h) RF_SBP2, (i) RF_SBP3, (j) RFRK_SBP1, (k) RFRK_SBP2, (l) RFRK_SBP3. Red fitting lines in triangles showed the trends.

**4 Discussion**

**4.1 Comparison of the GLM, RF, and their hybrid interpolators using ILR data**

The range of applicability of this study is limited to independent modelling. However, the study demonstrated the correlation of the raw data (sand, silt, and clay), and has confirmed that the currently used prediction models are suitable. For the assessment of independent validation, the RF revealed more accurate results, but with more bias than the GLM. The RK method improved the bias performance  for most models and the accuracy of the RF. Odeh et al. (1995)  indicated that RK was superior to the linear models, such as MLR, which was reflected in the prediction results for sand in our study. Scarpone et al. (2016) reported that as a hybrid interpolator, the RFRK outperformed the RF when making soil thickness predictions. We proved that RK was also suitable for compositional data and improved model performance when using an ILR transformation in the RF. In summary, the GLM and RF had both advantages and disadvantages when considering the trade-off between bias and accuracy. The difficulty with the use of the GLM is the need for a back-transformation. There is a need to present results on the original untransformed scale after conducting the analysis on a transformed level, which may produce spurious results (Lane, 2002). In our study, we compared the means of ILR transformed data and the original data. We proved the feasibility of the ILR transformation method, especially for meeting the requirements of compositional data. However, the accuracy of the GLM still needs to be improved, which may be because the transformed data did not follow a normal distribution. In addition, although the RF had the advantage of prediction accuracy, the limited interpretability of the consequences – a "black box" effect – made it difficult to modify the prediction bias because each tree from the model cannot be examined individually (Grimm et al., 2008). The ILR transformation before modeling increased the difficulty of interpretation for not only the predicted values on the ILR- scale but also the residuals. Moreover, the back-transformation of the optimal estimate of log-ratio variables does not generate the optimal estimation of compositional data (Lark and Bishop, 2007), which should also be considered. Multivariate methods, such as the multivariate RF, can be combined with a log-ratio transformation and hybrid interpolation, enabling the cross correlations among ILR coordinates to be better interpreted.

**4.2 Comparison of three SBP balances in the ILR transformation method**

The results of GLM and GLMRK should not depend on the ILR basis being chosen, which has been proved by previous studies on the use of linear models and kriging for compositional data (Pawlowsky-Glahn et al, 2015). However, the GLM model used the "glmStepAIC" algorithm (i.e., a stepwise regression) to select the best combination of environmental covariables for each ILR component. Therefore, the variable inputs were different for these ILR data, and further impact the accuracy assessment and prediction maps.

The comparison of the three SBP balances showed that the indicators of ME and RMSE performed better when using SBP1 for ILR transformed data , which may be interpreted as the distributions of the ILR1 and ILR2

of SBP1 being more symmetric (Fig. 2b). In contrast, the performance of SBP2 was worse than that of SBP1

and SBP3 because the ILR_1 component, including all the soil PSF information , was left-skewed (Fig. 2c). This result was especially apparent for the GLM and GLMRK, because the  data in the linear model needs to be normally distributed (Lane, 2002).

The interpretation of the negligible difference among the three SBP balances was  presented in a biplot of soil

PSF sampling data (Fig. 3b), which revealed a triangular shape. This could be interpreted as the three soil PSFs having a mixed pattern, with each component  dominated by the components in one cluster (Tolosana-Delgado et al., 2005). Although the silt component dominated the soil PSFs  (Fig. 2a), sand and clay also played important roles in the compositional data . Therefore, taking either the most abundant component of the compositional data as the denominator (Martins et al., 2016) or the first component of the permutations did not provide convincing evidence. Using the most abundant component of the compositional data as the primary component of the alterations, i.e., SBP2, resulted in a relatively poor performance compared to the other SBPs . Thus, we recommend using other parts that are not the most abundant as the first component of permutations , which in this case resulted in a uniform distribution on the biplot diagram, with a cluster at about 120 ° (Fig. 3b). Furthermore, the choice of balance is the key to improving model accuracy, as shown by the result of the RFRK-SBP3 model (Fig. 4). We also fitted the biplots using a random sampling test (70 % of the soil  data  randomly sampled), and the distributions (angle) of these graphs  were almost the same (Fig. S3.1). Multiple data sets should be considered in further studies to verify if this is a general feature of soil PSF samples or if it was produced from our data set.

The weighting problem was not considered in this study, because the ILR method can be qualified as an unweighted log-ratio transformation, giving all parts the same weight for both the definition of the total variance and the reduction of dimension. This may enlarge the ratios generated from the rare parts , which would dominate the analysis (Greenacre and

Lewi, 2009). The pairwise log-ratio can be used to set weights by their proportions when there is no additional knowledge about the component measurement errors (Greenacre, 2019). Nevertheless, all three parts of the soil PSF data dominated the biplot diagram, without the influence of rare elements and with no redundancy; thus, none of the shortcomings mentioned above  were apparent. Accuracy assessments using a pairwise logratio transformation require further study in the future.

**4.3 Limitations**

In this work, we used ILR transformation to demonstrate the correlation of soil PSF data, and different balances were also compared. However, these models were predicted separately for each ILR component (ILR1 and ILR2), which were suboptimal because they cannot further consider the cross correlations among ILR coordinates. In our pervious study, we have used compositional kriging (CK) for the spatial prediction of soil PSFs (Wang and Shi, 2017), and the cross correlations of

ILRs can be taken into account using CK. Although it is optimal, it cannot consider different balances of ILR, nor can it be
combined with the hybrid interpolator (e.g., RK). Moreover, predicting each ILR component separately was a more suitable
approach for the spatial prediction models currently used (such as the GLM and RF). Therefore, more alternative spatial
prediction models combined with interpretation of ILR balances for compositional data should be considered in the future. For
example, CK and high accuracy surface modelling (HASM; Yue et al., 2016) can be applied for small scale study areas. For
large scale study areas, multivariate RF (Segal and Xiao, 2011) can be combined with a log-ratio transformation and hybrid
interpolation, enabling the cross correlations among ILR coordinates to be better interpreted.

**5 Conclusions**

We evaluated and compared the performance of the GLM, RF, and their hybrid pattern (i.e., GLMRK and RFRK) using
different  balances of ILR transformed data. The bias of the GLM was lower than that of the RF; however, the
accuracy of the GLM was relatively low. More discrete distributions and broader ranges of prediction value distributions
were produced from GLMs in the USDA soil texture triangles. In other words, different data sets were generated from the use
of the GLM and RF, with unbiased and inaccurate predictions for the GLM and biased and more accurate predictions for the
RF.

The hybrid pattern of GLM and RF (i.e., RK) was found to be the best solution
because it produced a relatively high prediction accuracy and strong
correlations with ECs, providing more details about the soil sampling points (hot spots and cold spots) compared with only the
regression model. However, the non-normal distribution of ILR transformed data, and the "black box" effect of the RF
algorithm were drawbacks in the use of the GLMRK and RFRK.

For the different SBP balances , the three SBP-based data generated slightly different distributions.
, but no pattern was apparent. This could be explained by
the angle of the biplot diagram, with three rays of soil PSF components clustered into three modes, and each part
dominating its cluster. Using the most abundant component of the compositional data as the first
component of the permutations was not considered the right choice because SBP2 produced the worst performance
Instead, we recommend using other parts that are not the most abundant as the first
component of permutations , which in this case resulted in a uniform distribution on the
biplot diagram, with a cluster at about 120 °. To consider the general features of soil
PSF compositional data, multiple soil PSF data sets should be considered and compared in the future. This study
can provide a reference for the spatial simulation of soil PSFs combined with ECs at the regional
scale, and how to choose the balances of ILR transformed data.

[revised manuscript text omitted]

Pawlowsky-Glahn V, Egozcue JJ, Tolosana-Delgado R.: Modeling and analysis of compositional data. John Wiley & Sons, Ltd, 2015.

R Development Core Team: R: A language and environment for statistical computing, in, R Foundation for Statistical Computing, Vienna, Austria, 2019.

Scarpone, C., Schmidt, M. G., Bulmer, C. E., and Knudby, A.: Modelling soil thickness in the critical zone for Southern British

Columbia, Geoderma, 282, 59-69, https://doi.org/10.1016/j.geoderma.2016.07.012, 2016.

Segal, M. and Xiao, Y. Y.: Multivariate random forests,Wiley Interdisciplinary Reviews-Data Mining and Knowledge Discovery, 1, 80–87, https://doi.org/10.1002/widm.12, 2011.

Song, X.-D., Brus, D. J., Liu, F., Li, D.-C., Zhao, Y.-G., Yang, J.-L., and Zhang, G.-L.: Mapping soil organic carbon content by geographically weighted regression: A case study in the Heihe River Basin, China, Geoderma, 261, 11-22, https://doi.org/10.1016/j.geoderma.2015.06.024, 2016.

Talska, R., Menafoglio, A., Machalova, J., Hron, K., and Fiserova, E.: Compositional regression with functional response, Computational Statistics & Data Analysis, 123, 66-85, 10.1016/j.csda.2018.01.018, 2018.

Tolosana-Delgado, R., Otero, N., Pawlowsky-Glahn, V., and Soler, A.: Latent compositional factors in the Llobregat River Basin (Spain) hydrogeochemistry, Mathematical Geology, 37, 681-702, https://doi.org/10.1007/s11004-005-7375-7, 2005.

Venables, W. N., and Dichmont, C. M.: GLMs, GAMs and GLMMs: an overview of theory for applications in fisheries research, Fisheries Research, 70, 319-337, https://doi.org/10.1016/j.fishres.2004.08.011, 2004.

Walvoort, D. J. J., and de Gruijter, J. J.: Compositional Kriging: A spatial interpolation method for compositional data, Mathematical Geology, 33, 951-966, https://doi.org/10.1023/a:1012250107121, 2001.

Wang, Z., and Shi, W. J.: Mapping soil particle-size fractions: A comparison of compositional kriging and log-ratio kriging, J. Hydrol., 546, 526-541, https://doi.org/10.1016/j.jhydrol.2017.01.029, 2017.

Wang, Z., and Shi, W. J.: Robust variogram estimation combined with isometric log-ratio transformation for improved accuracy of soil particle-size fraction mapping, Geoderma, 324, 56-66, https://doi.org/10.1016/j.geoderma.2018.03.007, 2018.

Yang, R.-M., Zhang, G.-L., Liu, F., Lu, Y.-Y., Yang, F., Yang, F., Yang, M., Zhao, Y.-G., and Li, D.-C.: Comparison of boosted regression tree and random forest models for mapping topsoil organic carbon concentration in an alpine ecosystem, Ecological Indicators, 60, 870-878, https://doi.org/10.1016/j.ecolind.2015.08.036, 2016.

Yi, C., Li, D., Zhang, G., Zhao, Y., Yang, J., Liu, F., and Song, X.: Criteria for partition of soil thickness and case studies, Acta Pedologica Sinica, 52, 220-227, 2015.

Yue, T., Liu, Y., Zhao, M., Du, Z., and Zhao, N.: A fundamental theorem of Earth's surface modelling, Environ. Earth Sci., 75, 751, https://doi.org/10.1007/s12665-016-5310-5, 2016.

Yue, T., Zhao, N., Liu, Y., Wang, Y., Zhang, B., Du, Z., Fan, Z., Shi, W., Chen, C., Zhao, M., Song, D., Wang, S., Song, Y., Yan, C., Li, Q., Sun, X., Zhang, L., Tian, Y., Wang, W., Wang, Y. a., Ma, S., Huang, H., Lu, Y., Wang, Q., Wang, C., Wang, Y., Lu, M., Zhou, W., Liu, Y., Wang, Z., Bao, Z., Zhao, M., Zhao, Y., Rao, Y., Naseer, U., Fan, B., Li, S., Yang, Y., and Wilson, J. P.: A fundamental theorem for eco-environmental surface modelling and its applications, Science China-Earth Sciences, 63, 1092-1112, https://doi.org/10.1007/s11430-019-9594-3, 2020.

Zhang, M., Shi, W., and Xu, Z.: Systematic comparison of five machine-learning models in classification and interpolation of soil particle size fractions using different transformed data, Hydrol. Earth Syst. Sci., 24, 2505-2526, https://doi.org/10.5194/hess-24-2505-2020, 2020.